# Pure and Spurious Critical Points: a Geometric Study of Linear Networks

**Matthew Trager**[*]
New York University

**Kathlén Kohn**[*]
KTH Stockholm

**Joan Bruna**
New York University

## Abstract

The critical locus of the loss function of a neural network is determined by the geometry of the functional space and by the parameterization of this space by the network's weights. We introduce a natural distinction between *pure* critical points, which only depend on the functional space, and *spurious* critical points, which arise from the parameterization. We apply this perspective to revisit and extend the literature on the loss function of linear neural networks. For this type of network, the functional space is either the set of all linear maps from input to output space, or a *determinantal variety*, *i.e.*, a set of linear maps with bounded rank. We use geometric properties of determinantal varieties to derive new results on the landscape of linear networks with different loss functions and different parameterizations. Our analysis clearly illustrates that the absence of "bad" local minima in the loss landscape of linear networks is due to two distinct phenomena that apply in different settings: it is true for arbitrary smooth convex losses in the case of architectures that can express all linear maps ("filling architectures") but it holds only for the quadratic loss when the functional space is a determinantal variety ("non-filling architectures"). Without any assumption on the architecture, smooth convex losses may lead to landscapes with many bad minima.

## 1 Introduction

A fundamental goal in the theory of deep learning is to explain why the optimization of the non-convex loss function of a neural network does not seem to be affected by the presence of non-global local minima. Many papers have addressed this issue by studying the *landscape* of the loss function (Baldi & Hornik, 1989; Choromanska et al., 2015; Kawaguchi, 2016; Venturi et al., 2018). These papers have shown that, in certain situations, any local minimum for the loss is in fact always a global minimum. Unfortunately, it is also known that this property does not apply in more general realistic settings (Yun et al., 2018; Venturi et al., 2018). More recently, researchers have begun to search for explanations based on the *dynamics* of optimization. For example, in certain limit situations, the gradient flow of over-parameterized networks will avoid local minimizers (Chizat & Bach, 2018; Mei et al., 2018). We believe however that the study of the *static* properties of the loss function (the structure of its critical locus) is not settled. Even in the case of *linear networks*, the existing literature paints a purely analytical picture of the loss, and provides no sort of explanation as to "why" such architectures exhibit no bad local minima. A complete understanding of the critical locus should be a prerequisite for investigating the dynamics of the optimization.

The goal of this paper is to revisit the loss function of neural networks from a geometric perspective, focusing on the relationship between the functional space of the network and its parameterization. In particular, we view the loss as a composition

$$\{\text{parameter space}\} \overset{\mu}{\to} \{\text{functional space}\} \overset{\ell}{\to} \mathbb{R}.$$

In this setting, the function $\ell$ is almost always convex, however the composition $L = \ell \circ \mu$ is not. Critical points for $L$ can in fact arise for two distinct reasons: either because we are applying $\ell$ to a non-convex functional space, or because the parameterizing map $\mu$ is locally degenerate. We distinguish these two types of critical points by referring to them, respectively, as *pure* and *spurious*.

---

[*]Equal contribution.

Table 1: Bad local minima in loss landscapes for linear networks

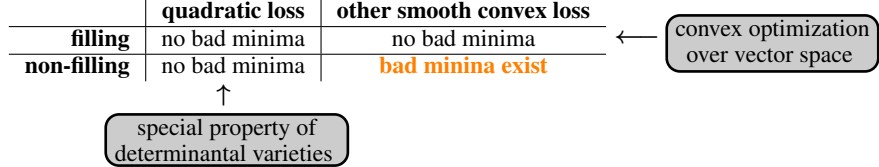

|  | quadratic loss | other smooth convex loss |
|---|---|---|
| **filling** | no bad minima | no bad minima |
| **non-filling** | no bad minima | **bad minina exist** |

Intuitively, pure critical points actually reflect the geometry of the functional space associated with the network, while spurious critical points arise as "artifacts" from the parameterization. After defining pure and critical points for arbitrary networks, we investigate in detail the classification of critical points in the case of linear networks. The functional space for such networks can be identified with a family of linear maps, and we can describe its geometry using algebraic tools. Many of our statements rely on a careful analysis of the differential of the matrix multiplication map. In particular, we prove that *non-global local minima are necessarily pure* critical points for convex losses, which means that many properties of the loss landscape can be read from the functional space. On the other hand, we emphasize that even for linear networks it is possible to find many smooth convex losses with non-global local minima. This happens when the functional space is a *determinantal variety*, *i.e.*, a (non-smooth and non-convex) family of matrices with bounded rank. In this setting, the absence of non-global minima actually holds in the particular case of the quadratic loss, because of very special geometric properties of determinantal varieties that we discuss.

**Related Work.** Baldi & Hornik (1989) first proved the absence of non-global ("bad") local minima for linear networks with one hidden layer (autoencoders). Their result was generalized to the case of deep linear networks by Kawaguchi (2016). Many papers have since then studied the loss landscape of linear networks under different assumptions (Hardt & Ma, 2016; Yun et al., 2017; Zhou & Liang, 2017; Laurent & von Brecht, 2017; Lu & Kawaguchi, 2017; Zhang, 2019). In particular, Laurent & von Brecht (2017) showed that linear networks with "no bottlenecks" have no bad local minima for arbitrary smooth loss functions. Lu & Kawaguchi (2017) and Zhang (2019) argued that "depth does not create local minima", meaning that the absence of local minima of deep linear networks is implied by the same property of shallow linear networks. Our study of pure and spurious critical points can be used as a framework for explaining all these results in a unified way. The *optimization dynamics* of linear networks are also an active area of research (Arora et al., 2019; 2018), and our analysis of the landscape in function space sets the stage for studying gradient dynamics on determinantal varieties, as in Bah et al. (2019). Our work is also closely related to objects of study in applied algebraic geometry, particularly *determinantal varieties* and *ED discriminants* (Draisma et al., 2013; Ottaviani et al., 2013). Finally, we mention other recent works that study neural networks using algebraic-geometric tools (Mehta et al., 2018; Kileel et al., 2019; Jaffali & Oeding, 2019).

**Main contributions.**

- We introduce a natural distinction between "pure" and "spurious" critical points for the loss function of networks. These notions provide an intuitive and useful language for studying a central aspect in the theory of neural networks, namely the (over)parameterization of the functional space and its effect on the optimization landscape. While most of the paper focuses on linear networks, this viewpoint applies to more general settings as well (see also our discussion in Appendix A.3).

- We study the pure and critical locus for linear networks and arbitrary loss functions. We show that non-global local minima are always pure for convex losses, unifying many known properties on the landscape of linear networks.

- We explain that the absence of "bad" local minima in the loss landscape of linear networks is due to two distinct phenomena and does not hold in general: it is true for arbitrary smooth convex losses in the case of architectures that can express all linear maps ("filling architectures") and it holds for the quadratic loss when the functional space is a determinantal variety ("non-filling architectures"). Without any assumption on the architecture, smooth convex losses may lead to many local minima. See Table 1.

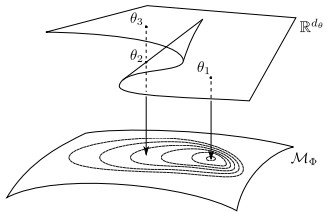

Figure 1: Pure and spurious critical points: $\theta_1$ is a pure critical point, while $\theta_2$ is a spurious critical point (the level curves on the manifold $\mathcal{M}_\Phi$ describe the landscape in functional space). Note that $\theta_3$ is mapped to the same function as $\theta_2$, but it is not a critical point.

- We provide a precise description of the number of topologically connected components of the set of global minima. This relates to recent work on "mode connectivity" in loss landscapes of neural networks (Garipov et al., 2018).

- We spell out connections between the loss landscape and classical geometric objects such as caustics and ED discriminants. We believe that these concepts may be useful in the study of more general functional spaces.

**Differential notation.** Our functional spaces will be manifolds with singularities, so we will make use of elementary notions from differential geometry. If $\mathcal{M}$ and $\mathcal{N}$ are manifolds and $g : \mathcal{M} \to \mathcal{N}$ is a smooth map, then we write $dg(x)$ for the differential of $g$ at the point $x$. This means that $dg(x) : T_x \mathcal{M} \to T_{g(x)} \mathcal{N}$ is the first order linear approximation of $g$ at the point $x \in \mathcal{M}$. If $\mathcal{M}$ and $\mathcal{N}$ have singularities, then the same definitions apply if we restrict $g$ to smooth points in $\mathcal{M}$ whose image is also smooth in $\mathcal{N}$. For most of our analysis, manifolds will be embedded in Euclidean spaces, say $\mathcal{M} \subset \mathbb{R}^m$ and $\mathcal{N} \subset \mathbb{R}^n$, so we can view the tangent spaces $T_x \mathcal{M}$ and $T_{g(x)} \mathcal{N}$ as also embedded in $\mathbb{R}^m$ and $\mathbb{R}^n$. When $\mathcal{N} = \mathbb{R}$, the critical locus of a map $g : \mathcal{M} \to \mathbb{R}$ is defined as $Crit(g) = \{x \in Smooth(\mathcal{M}) \mid dg(x) = 0\}$.

## 2 PRELIMINARIES

### 2.1 PURE AND SPURIOUS CRITICAL POINTS

A neural network (or any general "parametric learning model") is defined by a continuous mapping $\Phi : \mathbb{R}^{d_\theta} \times \mathbb{R}^{d_x} \to \mathbb{R}^{d_y}$ that associates an input vector $x \in \mathbb{R}^{d_x}$ and a set of parameters $\theta \in \mathbb{R}^{d_\theta}$ to an output vector $y = \Phi(\theta, x) \in \mathbb{R}^{d_y}$. In other words, $\Phi$ determines a family of continuous functions parameterized by $\theta \in \mathbb{R}^{d_\theta}$:

$$\mathcal{M}_\Phi = \{f_\theta : \mathbb{R}^{d_x} \to \mathbb{R}^{d_y} \mid f_\theta = \Phi(\theta, \cdot)\} \subset C(\mathbb{R}^{d_x}, \mathbb{R}^{d_y}).$$

Even though $\mathcal{M}_\Phi$ is naturally embedded in an infinite-dimensional functional space, it is itself finite dimensional. In fact, if the mapping $\Phi$ is smooth, then $\mathcal{M}_\Phi$ is a finite-dimensional manifold with singularities, and its intrinsic dimension is upper bounded by $d_\theta$. It is also important to note that neural networks are often *non-identifiable models*, which means that different parameters can represent the same function (*i.e.*, $f_\theta = f_{\theta'}$ does not imply $\theta = \theta'$). The manifold $\mathcal{M}_\Phi$ is sometimes known as a *neuromanifold* (Amari, 2016). We now consider a general loss function of the form $L = \ell \circ \mu$, where $\mu : \mathbb{R}^{d_\theta} \to \mathcal{M}_\Phi$ is the (over)parameterization of $\mathcal{M}_\Phi$ by $\theta$ and $\ell$ is a functional defined on a subset of $C(\mathbb{R}^{d_x}, \mathbb{R}^{d_y})$ containing $\mathcal{M}_\Phi$:[1]

$$L : \mathbb{R}^{d_\theta} \xrightarrow{\mu} \mathcal{M}_\Phi \xrightarrow{\ell|_{\mathcal{M}_\Phi}} \mathbb{R}. \tag{1}$$

**Definition 1.** A critical point $\theta^* \in Crit(L)$ is a *pure critical point* if $\mu(\theta^*)$ is a critical point for the restriction $\ell|_{\mathcal{M}_\Phi}$ (note that this implicitly requires $\mu(\theta^*)$ to be a smooth point of $\mathcal{M}_\Phi$). If $\theta^* \in Crit(L)$ but $\mu(\theta^*) \notin Crit(\ell|_{\mathcal{M}_\Phi})$, we say that $\theta^*$ is a *spurious critical point*.

---

[1]This setting applies to both the empirical loss and the population loss.

It is clear from this definition that pure critical points reflect the geometry of the functional space, while spurious critical points do not have an intrinsic functional interpretation. For example, if $\theta^* \in Crit(L)$ is a spurious critical point, then it may be possible to find another parameter $\theta'$ that represents the same function $f_{\theta^*} = f_{\theta'}$ and is not a critical point for $L$ (see Figure 1). In contrast, if $\theta^*$ is a pure critical point, then *all* parameters $\theta'$ such that $\mu(\theta') = \mu(\theta^*)$ are automatically in $Crit(L)$, simply because $dL(\theta') = d\ell|_{\mathcal{M}_\Phi}(\mu(\theta')) \circ d\mu(\theta')$. This will motivate us to study the *fiber* $\{\theta \mid \mu(\theta) = f\}$ of all parameters mapped to the same function $f$ (particularly when the function $f$ is a critical point of $\ell|_{\mathcal{M}_\Phi}$).

We note that a sufficient condition for $\theta^* \in Crit(L)$ to be a pure critical point is that the differential $d\mu(\theta^*)$ at $\theta^*$ has maximal rank (namely $\dim \mathcal{M}_\Phi$), *i.e.*, that $\mu$ is locally a *submersion* at $\theta^*$. Indeed, we have in this case

$$0 = dL(\theta^*) = d\ell|_{\mathcal{M}_\Phi}(\mu(\theta^*)) \circ d\mu(\theta^*) \Rightarrow d\ell|_{\mathcal{M}_\Phi}(\mu(\theta^*)) = 0,$$

so $\mu(\theta^*)$ is critical for the restriction of $\ell$ to $\mathcal{M}_\Phi$. We also point out a special situation when $\mathcal{M}_\Phi$ is a *convex set* (as a subset of $C(\mathbb{R}^{d_x}, \mathbb{R}^{d_y})$) and $\ell$ is a *smooth convex functional*. In this case, the only critical points of $\ell|_{\mathcal{M}_\Phi}$ are global minima, so we deduce that any critical point of $L = \ell \circ \mu$ is either a global minimum or a spurious critical point. The following simple observation gives a sufficient condition for critical points to be *saddles* (*i.e.*, they are not local minima or local maxima).

**Lemma 2.** *Let $\theta^* \in Crit(L)$ be a (necessarily spurious) critical point with the following property: for any open neighborhood $U$ of $\theta$, there exists $\theta'$ in $U$ such that $\mu(\theta') = \mu(\theta)$ and $\theta' \notin Crit(L)$. Then $\theta^*$ is a saddle for $L$.*

*Proof.* Assume that $\theta^*$ is a local minimum (the reasoning is analogous if $\theta^*$ is a local maximum). This means that there exists a neighborhood $U$ of $\theta^*$ such that $L(\theta) \geq L(\theta^*)$ for all $\theta \in U$. In particular, if $\theta' \in U$ is such that $\mu(\theta') = \mu(\theta)$, then $\theta'$ must also be a local minimum. This contradicts $\theta' \notin Crit(L)$. $\square$

This general discussion on pure and spurious critical points applies to any smooth network map $\Phi$ (with possible extensions to the case of piece-wise smooth mappings), and we believe that the distinction can be a useful tool in the study of the optimization landscape of general networks. In the remaining part of the paper, we use this perspective for an in-depth study of the critical points of *linear networks*. For this type of network, the functional set $\mathcal{M}_\Phi$ can be embedded in a finite dimensional ambient space, namely the space of all linear maps $\mathbb{R}^{d_x} \to \mathbb{R}^{d_y}$. Furthermore, $\mathcal{M}_\Phi$ is an algebraic variety (a manifold that can have singularities and that can be described by algebraic equations). We will use basic tools from algebraic geometry to provide a complete description of pure and spurious critical points, and to prove new results on the landscape of linear networks.

## 2.2 LINEAR NETWORKS AND DETERMINANTAL VARIETIES

A *linear network* is a map $\Phi : \mathbb{R}^{d_\theta} \times \mathbb{R}^{d_x} \to \mathbb{R}^{d_y}$ of the form

$$\Phi(\theta, x) = W_h \ldots W_1 x, \qquad \theta = (W_h, \ldots, W_1) \in \mathbb{R}^{d_\theta}, \tag{2}$$

where $W_i \in \mathbb{R}^{d_i \times d_{i-1}}$ are matrices (so $d_0 = d_x$, $d_h = d_y$, and $d_\theta = d_0 d_1 + d_1 d_2 + \ldots + d_{h-1} d_h$). The functional space is in this case a subset of the space of all linear maps $\mathbb{R}^{d_0} \to \mathbb{R}^{d_h}$. As in (1), we can decompose a loss function $L$ for a linear network $\Phi$ as

$$
\begin{array}{ccccc}
\mathbb{R}^{d_h \times d_{h-1}} \times \ldots \times \mathbb{R}^{d_1 \times d_0} & \xrightarrow{\mu_{\boldsymbol{d}}} & \mathbb{R}^{d_h \times d_0} & \xrightarrow{\ell} & \mathbb{R} \\
(W_h, \ldots, W_1) & \longmapsto & \overline{W} = W_h \ldots W_1 & \longmapsto & \ell(\overline{W}).
\end{array} \tag{3}
$$

Here $\mu_{\boldsymbol{d}}$ is the *matrix multiplication map* for the sequence of widths $\boldsymbol{d} = (d_h, \ldots, d_0)$, and $\ell$ is a functional on the space of $(d_h \times d_0)$-matrices. In practice, it is typically a functional that depends on the training data, e.g. $\ell(\overline{W}) = \|\overline{W}X - Y\|^2$ for fixed matrices $X, Y$.[2] Note that even if $\ell$ is a convex functional, the set $\mathcal{M}_\Phi$ will often not be a convex set. In fact, it is easy to see that the image of $\mu_{\boldsymbol{d}}$ is the space $\mathcal{M}_r$ of $(d_h \times d_0)$-matrices of rank at most $r = \min\{d_0, \ldots, d_h\}$. If $r < \min(d_0, d_h)$, this

---

[2]Our setting can also be applied when $\ell$ includes a regularizer term defined in function space, *e.g.*, $\ell(\overline{W}) = \|\overline{W}X - Y\|^2 + \lambda R(\overline{W})$.

set is known as a *determinantal variety*, a classical object of study in algebraic geometry (Harris, 1995). It is in fact an *algebraic variety*, *i.e.*, it is described by polynomial equations in the matrix entries (namely, it is the zero-set of all $(r + 1) \times (r + 1)$-minors), and it is well known that the dimension of $\mathcal{M}_r$ is $r(m + n - r)$. Furthermore, for $r > 0$, the variety $\mathcal{M}_r$ has many singularities: its singular locus is exactly $\mathcal{M}_{r-1} \subset \mathcal{M}_r$, the set of all matrices with rank strictly smaller than $r$. We refer the reader to Appendix A.1 for more details on determinantal varieties.

## 3    MAIN RESULTS

In this section, we investigate the critical locus $Crit(L)$ of general functions $L : \mathbb{R}^{d_\theta} \to \mathbb{R}$ of the form $L = \ell \circ \mu_{\boldsymbol{d}}$ where $\ell : \mathbb{R}^{d_h \times d_0} \to \mathbb{R}$ is a (often convex) smooth map, and $\mu_{\boldsymbol{d}}$ is the matrix multiplication map introduced in (3). By studying the differential of $\mu_{\boldsymbol{d}}$, we will characterize pure and spurious critical points of $L$. As previously noted, the image of $\mu_{\boldsymbol{d}}$ is $\mathcal{M}_r \subset \mathbb{R}^{d_h \times d_0}$ where $r = \min\{d_i\}$. In particular, we distinguish between two cases:

- We say that the map $\mu_{\boldsymbol{d}}$ is *filling* if $r = \min\{d_0, d_h\}$, so $\mathcal{M}_r = \mathbb{R}^{d_h \times d_0}$. In this case, the functional space is *smooth* and *convex*.

- We say that the map $\mu_{\boldsymbol{d}}$ is *non-filling* if $r < \min\{d_0, d_h\}$, so $\mathcal{M}_r \subsetneq \mathbb{R}^{d_h \times d_0}$ is a determinantal variety. In this case, the functional space is *non-smooth* and *non-convex*.

### 3.1    PROPERTIES OF THE MATRIX MULTIPLICATION MAP

We present some general results on the matrix multiplication map $\mu_{\boldsymbol{d}}$, which we will apply to linear networks in the next subsection. These facts may also be useful in other settings, for example, to study the piece-wise linear behavior of ReLU networks.

We begin by noting that the differential map of $\mu_{\boldsymbol{d}}$ can be written explicitly as

$$d\mu_{\boldsymbol{d}}(\theta)(\dot{W}_h, \ldots, \dot{W}_1) = \dot{W}_h W_{h-1} \ldots W_1 + W_h \dot{W}_{h-1} \ldots W_1 + \ldots + W_h \ldots W_2 \dot{W}_1. \quad (4)$$

Given a matrix $M \in \mathbb{R}^{m \times n}$, we denote by $Row(M) \subset \mathbb{R}^n$ and $Col(M) \subset \mathbb{R}^m$ the vector spaces spanned by the rows and columns of $M$, respectively. Writing $W_{>i} = W_h W_{h-1} \ldots W_{i+1}$ and $W_{<i} = W_{i-1} W_{i-1} \ldots W_1$, the image of $d\mu_{\boldsymbol{d}}(\theta)$ in (4) is

$$\mathbb{R}^{d_h} \otimes Row(W_{<h}) + \ldots + Col(W_{>i}) \otimes Row(W_{<i}) + \ldots + Col(W_{>1}) \otimes \mathbb{R}^{d_0}. \quad (5)$$

From this expression, we deduce the following useful fact.

**Lemma 3.** *The dimension of the image of the differential $d\mu_{\boldsymbol{d}}$ at $\theta = (W_h, \ldots, W_1)$ is given by*

$$\mathrm{rk}(d\mu_{\boldsymbol{d}}(\theta)) = \sum_{i=1}^{h} \mathrm{rk}(W_{>i}) \cdot \mathrm{rk}(W_{<i}) - \sum_{i=1}^{h-1} \mathrm{rk}(W_{>i}) \cdot \mathrm{rk}(W_{<i+1}),$$

*where we use the convention that $W_{<1} = I_{d_0}$, $W_{>h} = I_{d_h}$ are the identity matrices of size $d_0$, $d_h$.*

We can use Lemma 3 to characterize all cases when the differential $d\mu_{\boldsymbol{d}}$ at $\theta = (W_h, \ldots, W_1)$ has full rank (*i.e.*, when the matrix multiplication map is a local submersion onto $\mathcal{M}_r$).

**Theorem 4.** *Let $r = \min\{d_i\}$, $\theta = (W_h, \ldots, W_1)$, and $\overline{W} = \mu_{\boldsymbol{d}}(\theta)$.*

- *(Filling case) If $r = \min\{d_h, d_0\}$, the differential $d\mu_{\boldsymbol{d}}(\theta)$ has maximal rank equal to $\dim \mathcal{M}_r = d_h d_0$ if and only if, for every $i \in \{1, 2, \ldots, h - 1\}$, either $\mathrm{rk}(W_{>i}) = d_h$ or $\mathrm{rk}(W_{<i+1}) = d_0$ holds.*

- *(Non-filling case) If $r < \min\{d_h, d_0\}$, the differential $d\mu_{\boldsymbol{d}}(\theta)$ has maximal rank equal to $\dim \mathcal{M}_r = r(d_h + d_0 - r)$ if and only if $\mathrm{rk}(\overline{W}) = r$.*

*Furthermore, in both situations, if $\mathrm{rk}(\overline{W}) = e < r$, then the image of $d\mu_{\boldsymbol{d}}(\theta)$ always contains the tangent space $T_{\overline{W}} \mathcal{M}_e$ of $\mathcal{M}_e \subset \mathcal{M}_r$ at $\overline{W}$.*

We note that $d\mu_{\boldsymbol{d}}(\theta)$ has always maximal rank when $\mathrm{rk}(\overline{W}) = r = \min\{d_i\}$, however in the filling case it is possible to obtain a local submersion even when $\mathrm{rk}(\overline{W}) < r$ (see Example 19 in appendix). We next describe the *fiber* of the matrix multiplication map, that is, the set

$$\mu_{\boldsymbol{d}}^{-1}(\overline{W}) = \{(W_h, \ldots, W_1) \mid \overline{W} = W_h \ldots W_1, \ W_i \in \mathbb{R}^{d_i \times d_{i-1}}\}.$$

It will be convenient to refer to $\mu_{\boldsymbol{d}}^{-1}(\overline{W})$ as the set of $\boldsymbol{d}$-*factorizations* of $\overline{W}$. We are interested in understanding the structure of $\mu_{\boldsymbol{d}}^{-1}(\overline{W})$ since, as argued in Section 2.1, pure critical loci consist of fibers of "critical functions". The following result completely describes the *connectivity* of $\mu_{\boldsymbol{d}}^{-1}(\overline{W})$.

**Theorem 5.** *Let $r = \min\{d_i\}$. If $\mathrm{rk}(\overline{W}) = r$, then the set of $\boldsymbol{d}$-factorizations $\mu_{\boldsymbol{d}}^{-1}(\overline{W})$ of $\overline{W}$ has exactly $2^b$ path-connected components, where $b = \#\{i \mid d_i = r, \ 0 < i < h\}$. If $\mathrm{rk}(\overline{W}) < r$, then $\mu_{\boldsymbol{d}}^{-1}(\overline{W})$ is always path-connected.*

### 3.2 APPLICATION TO LINEAR NETWORKS

We now apply the general results from the previous subsection to study the critical locus $Crit(L)$ with $L = \ell \circ \mu_{\boldsymbol{d}}$, where $\ell$ is any smooth function. In the following, we always use $r = \min\{r_i\}$ and $\overline{W} = \mu_{\boldsymbol{d}}(\theta)$. The next two facts follow almost immediately from Theorem 4.

**Proposition 6.** *If $\theta$ is such that $d\mu_{\boldsymbol{d}}(\theta)$ has maximal rank (see Theorem 4), then $\theta \in Crit(L)$ if and only if $\overline{W} \in Crit(\ell|_{\mathcal{M}_r})$, and $\theta$ is a minimum (resp., saddle, maximum) for $L$ if and only if $\overline{W}$ is a minimum (resp., saddle, maximum) for $\ell|_{\mathcal{M}_r}$. If $\mathrm{rk}(\overline{W}) = r$ (which implies that $d\mu_{\boldsymbol{d}}(\theta)$ has maximal rank) and $\theta \in Crit(L)$, then all $\boldsymbol{d}$-factorizations of $\overline{W}$ also belong to $Crit(L)$.*

**Proposition 7.** *If $\theta \in Crit(L)$ with $\mathrm{rk}(\overline{W}) = e \leq r$, then $\overline{W} \in Crit(\ell|_{\mathcal{M}_e})$. In other words, if $\mathrm{rk}(\overline{W}) < r$, then $\theta \in Crit(L)$ implies that $\overline{W}$ is a critical point for the restriction of $\ell$ to a smaller determinantal variety $\mathcal{M}_e$ (which is in the singular locus of the functional space $\mathcal{M}_r$ in the non-filling case).*

Note that if $d_h = 1$, then either $\overline{W} = 0$ or $\mathrm{rk}(\overline{W}) = 1$, and in the latter case Proposition 7 implies that $\overline{W} \in Crit(\ell|_{\mathbb{R}^{d_0} \setminus \{0\}})$. If $\ell$ is convex, we immediately obtain that *all critical points* (not just local minima, as in Laurent & von Brecht (2017)) below a certain energy level are *global minima*.

**Corollary 8.** *Assume that $\ell$ is a smooth convex function and that $d_h = 1$. If $\theta \in Crit(L)$, then either $\overline{W} = \mu_{\boldsymbol{d}}(\theta) = 0$ or $\theta$ is* global minimum *for $L$.*

Proposition 7 shows that critical points for $L$ such that $\mathrm{rk}(\overline{W}) < r$ correspond to critical points for $\ell$ restricted to a smaller determinantal variety. Using Lemma 2, it is possible to show that these points are essentially always *saddles* for $L$.

**Proposition 9.** *Let $\theta \in Crit(L)$ be such that $\mathrm{rk}(\overline{W}) < r$, and assume that $d\ell(\overline{W}) \neq 0$. Then, for any neighborhood $U$ of $\theta$, there exists $\theta'$ in $U$ such that $\mu_{\boldsymbol{d}}(\theta') = \overline{W}$ but $\theta' \notin Crit(L)$. In particular, $\theta$ is a saddle point.*

**Proposition 10.** *Let $\ell$ be any smooth convex function, and let $L = \ell \circ \mu_{\boldsymbol{d}}$. If $\theta$ is a non-global local minimum for $L$, then necessarily $\mathrm{rk}(\overline{W}) = r$ (so $\theta$ is a* pure *critical point). In particular, $L$ has non-global minima if and only if $\ell|_{\mathcal{M}_r}$ has non-global minima.*

This statement succinctly explains many known facts on the landscape of linear networks. For example, we recover the main result from (Laurent & von Brecht, 2017), which states that when $\ell$ is a smooth convex function and $\mu_{\boldsymbol{d}}$ is filling ($r = \min\{d_h, d_0\}$), then all local minima for $L$ are global minima: indeed, this is because $\mathcal{M}_r = \mathbb{R}^{d_h \times d_0}$ is a linear space, so $\ell|_{\mathcal{M}_r}$ does not have non-global minima. On the other hand, when $\mu_{\boldsymbol{d}}$ is not filling, the functional space is not convex, and multiple local minima may exist even when $\ell$ is a convex function. We will in fact present many examples of smooth convex functions $\ell$ such that $L = \ell \circ \mu_{\boldsymbol{d}}$ has non-global local minima (see Figure 3). In the special case that $\ell$ is a quadratic loss (for any data distribution), then it is a remarkable fact that there are no non-global local minima even when $\mu_{\boldsymbol{d}}$ is not filling (Baldi & Hornik, 1989; Kawaguchi, 2016). In the next section, we will provide an intrinsic geometric justification for this property.

**Remark 11.** In Laurent & von Brecht (2017), the authors observe that their "structural hypothesis" (*i.e.*, for us, the fact that the network is filling) is a necessary assumption for their main result,

as otherwise critical points of $\ell$ might not lie in the functional space of the network. This last observation however does not imply the necessity of the filling assumption, and indeed in the case of the quadratic loss there are no local bad minima despite the fact $\mathcal{M}_r \subsetneq \mathbb{R}^{d_h \times d_0}$.

Finally, we conclude this section by pointing out that although the pure critical locus is determined by the geometry of the functional space, the "lift" from function space to parameter space is not completely trivial. In particular, there is always a large positive-dimensional set of critical parameters associated with a critical linear function $\overline{W}$ (all possible $\boldsymbol{d}$-factorizations of $\overline{W}$). More interestingly, this set may be topologically disconnected into a large number of components that are all functionally equivalent (see Theorem 5). This observation agrees with the folklore knowledge that neural networks can have *many disconnected valleys* where the loss function achieves the same value.

## 3.3 THE QUADRATIC LOSS

We now assume that $\ell : \mathbb{R}^{d_y \times d_x} \to \mathbb{R}$ is of the form $\ell(\overline{W}) = \|\overline{W}X - Y\|^2$, where $X \in \mathbb{R}^{d_x \times s}$ and $Y \in \mathbb{R}^{d_y \times s}$ are fixed data matrices. As mentioned above, it is known that $L = \ell \circ \mu_{\boldsymbol{d}}$ has no non-global local minima, even when $\mu_{\boldsymbol{d}}$ is non-filling (Baldi & Hornik, 1989; Kawaguchi, 2016). In this section, we discuss the intrinsic geometric reasons for this special behavior.

It is easy to relate the landscape of $L$ with the *Euclidean distance function from a determinantal variety* (or, equivalently, to the problem *low-rank matrix approximation*). Indeed, we know from Proposition 10 that $L$ has non-global local minima if and only if the same is true for $\ell|_{\mathcal{M}_r}$. Furthermore, assuming that $XX^T$ has full rank, we use its square root $P = (XX^T)^{1/2}$ as a positive definite matrix to derive

$$\|WX - Y\|^2 = \langle WX, WX \rangle_F - 2\langle WX, Y \rangle_F + \langle Y, Y \rangle_F$$
$$= \langle WP, WP \rangle_F - 2\langle WP, YX^TP^{-1} \rangle_F + \langle Y, Y \rangle_F$$
$$= \|WP - Q_0\|^2 + const.,$$

where $Q_0 = YX^TP^{-1}$ and *"const."* only depends on the data matrices $X$ and $Y$. Hence, minimizing $\ell(W)$ is equivalent to minimizing $\|WP - Q_0\|^2$. Since the bijection $\mathcal{M}_r \to \mathcal{M}_r, W \mapsto WP$ is also a bijective on the tangent spaces, it provides a one-to-one correspondence from the critical points of $\min_{W \in \mathcal{M}_r} \|WP - Q_0\|^2$ to the critical points of $\min_{W \in \mathcal{M}_r} \|W - Q_0\|^2$. All in all, studying the critical points of $\ell|_{\mathcal{M}_r}$ is equivalent to studying the critical points of the function $h_{Q_0}(W) = \|W - Q_0\|^2$ where $W$ is restricted to the determinantal variety $\mathcal{M}_r$.

The function $h_{Q_0}(W)$ is described by following generalization of the classical Eckart-Young Theorem. The formulation we prove is an extension of Example 2.3 in Draisma & Horobet (2014) and Theorem 2.9 in Ottaviani et al. (2013). We consider a fixed matrix $Q_0 \in \mathbb{R}^{d_y \times d_x}$ and a singular value decomposition (SVD) $Q_0 = U\Sigma V^T$, where we assume $\Sigma \in \mathbb{R}^{d_y \times d_x}$ has decreasing diagonal entries $\sigma_1, \ldots, \sigma_m$, with $m = \min(d_y, d_x)$. For any $\mathcal{I} \subset \{1, 2, \ldots, m\}$ we write $\Sigma_{\mathcal{I}} \in \mathbb{R}^{d_y \times d_x}$ for the diagonal matrix with entries $\sigma_{\mathcal{I},1}, \ldots, \sigma_{\mathcal{I},m}$ where $\sigma_{\mathcal{I},i} = \sigma_i$ if $i \in \mathcal{I}$ and $\sigma_{\mathcal{I},i} = 0$ otherwise.

**Theorem 12.** *If the singular values of $Q_0$ are pairwise distinct and positive, $h_{Q_0}|_{\mathcal{M}_r}$ has exactly $\binom{m}{r}$ critical points, namely the matrices $Q_{\mathcal{I}} = U\Sigma_{\mathcal{I}}V^T$ with $\#(\mathcal{I}) = r$. Moreover, its unique local and global minimum is $Q_{\{1,\ldots,r\}}$. More precisely, the index of $Q_{\mathcal{I}}$ as a critical point of $h_{Q_0}|_{\mathcal{M}_r}$ (i.e., the number of negative eigenvalues of the Hessian matrix for any local parameterization) is*

$$\mathrm{index}(Q_{\mathcal{I}}) = \#\{(j,i) \in \mathcal{I} \times \mathcal{I}^c \mid j > i\}, \qquad \text{where } \mathcal{I}^c = \{1, \ldots, m\} \setminus \mathcal{I}.$$

In the appendix we present a more general version of this statement without the assumption that the singular values of $Q_0$ are pairwise distinct and positive. The surprising aspect of this result is that the structure of the critical points is the same for almost *all* choices of $Q_0$. We want to emphasize that this is a special behavior of determinantal varieties with respect to the Euclidean distance, and the situation changes drastically if we apply even infinitesimal changes to the quadratic loss function. More precisely, any linear perturbation of the Euclidean norm will result in a totally different landscape, as the following example shows (more details are given in Appendix A.2).

**Example 13.** Let us consider the variety $\mathcal{M}_1 \subseteq \mathbb{R}^{3 \times 3}$ of rank-one $(3 \times 3)$-matrices. By Theorem 12, for almost all $Q_0$, the function $h_{Q_0}|_{\mathcal{M}_1}$ has three (real) critical points. Applying a linear change of coordinates to $\mathbb{R}^{d_y \times d_x} \cong \mathbb{R}^{d_y d_x}$ yields a different quadratic loss $\tilde{h}_{Q_0}$. Using tools from algebraic

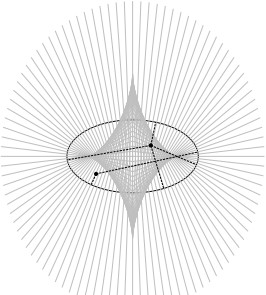 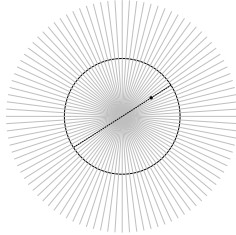

Figure 2: *Left:* If $\mathcal{V} \subset \mathbb{R}^2$ is an ellipse, the distance function $h_u(p) = \|p - u\|^2$ restricted to $\mathcal{V}$ generally has 2 or 4 real critical points, depending on whether $u$ lies inside or outside the diamond-shaped region bounded by the caustic curve. *Right:* If $\mathcal{V} \subset \mathbb{R}^2$ is a circle, then the caustic curve degenerates to a point and the distance function generically has always 2 real critical points.

geometry, it is possible to show that for almost all linear coordinate changes (an open dense set), the function $\tilde{h}_{Q_0}|_{\mathcal{M}_1}$ has 39 critical points over the complex numbers.[3] The number of real critical points however varies, depending on whether $Q_0$ belongs to different open regions separated by a *caustic hypersurface* in $\mathbb{R}^{3 \times 3}$. Furthermore, the number of local minima varies as well; in particular, it is no longer true that all $Q_0$ admit a unique local minimum. Figure 3 presents some simple computational experiments illustrating this behavior.

For all determinantal varieties, the situation is similar to the description in Example 13. More generally, given an algebraic variety $\mathcal{V} \subset \mathbb{R}^n$ and a point $u \in \mathbb{R}^n$, the number of (real) critical points of the distance function $h_u(p) = \|p - u\|^2$ restricted to $\mathcal{V}$ is usually not constant as $u$ varies: the behavior changes when $u$ crosses the *caustic hypersurface*, or *ED (Euclidean distance) discriminant*, of $\mathcal{V}$; see Figure 2. In the case of determinantal varieties with the standard Euclidean distance, this caustic hypersurface (more precisely its real locus) degenerates to a set of codimension 2, which does not partition the space into different regions. This is analogous to the case of the circle in Figure 2.

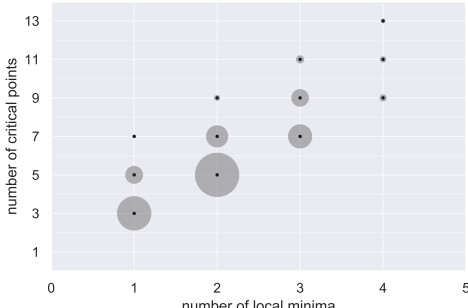

Figure 3: Real critical points and local minima for random choices of $\tilde{h}_{Q_0}|_{\mathcal{M}_1}$ as defined in Example 13. The size of each disk is proportional to the number of instances we found with that number of critical points and local minima. This shows that linear networks with a convex loss may indeed have multiple non-global local minima. More details in Appendix A.2 (Table 2 and Experiment 1).

### 3.4 USING DIFFERENT PARAMETERIZATIONS: NORMALIZED NETWORKS

In the simple linear network model (2), the functional space $\mathcal{M}_r \subset \mathbb{R}^{d_h \times d_0}$ is parameterized using the matrix multiplication map $\mu_{\boldsymbol{d}}$. On the other hand, one can envision many variations of this model that are network architectures with the same functional space but parameterized differently. Examples include linear networks with skip connections, or convolutional linear networks. In this

---

[3]This means that the algebraic equations corresponding to the vanishing of the differential have exactly 39 complex solutions.

subsection, we take a look at a model for *normalized linear networks*: these are maps of the form

$$\Psi(\theta, x) = W_h \frac{W_{h-1}}{\|W_{h-1}\|} \cdots \frac{W_1}{\|W_1\|} x, \qquad \theta = (W_h, \ldots, W_1), \tag{6}$$

where $W_i \in \mathbb{R}^{d_i \times d_{i-1}}$ as before. This is a simple model for different types of weight normalization schemes often used in practice. It is easy to see that the difference between (6) and our previous linear network lies only in the parameterization of linear maps, since for normalized networks the matrix multiplication map is replaced by

$$\nu_{\boldsymbol{d}} : \Omega \to \mathbb{R}^{d_h \times d_0}, \quad (W_h, \ldots, W_1) \mapsto \overline{W} = W_h \frac{W_{h-1}}{\|W_{h-1}\|} \cdots \frac{W_1}{\|W_1\|},$$

where $\Omega = \{(W_h, \ldots, W_1) \,|\, W_i \neq 0, i = 1, \ldots, h-1\} \subset \mathbb{R}^{d_\theta}$. According to our definitions, if $L = \ell \circ \mu_{\boldsymbol{d}}$ and $L' = \ell \circ \nu_{\boldsymbol{d}}$ are losses respectively for linear networks and normalized linear networks, then the pure critical loci of $L$ and $L'$ will correspond to each other (since these only depend on the functional space), but a priori the spurious critical loci induced by the two parameterizations may be different. In this particular setting, however, we show that this is not the case: the new paramerization effectively does not introduce different critical points, and in fact makes the critical locus slightly smaller.

**Proposition 14.** *If $L' = \ell \circ \nu_{\boldsymbol{d}}$ and $L = \ell \circ \mu_{\boldsymbol{d}}$, then the critical locus $Crit(L')$ is in "correspondence" with $Crit(L) \cap \Omega$, meaning that*

$$\{\nu_{\boldsymbol{d}}(\theta') \,|\, \theta' \in Crit(L')\} = \{\mu_{\boldsymbol{d}}(\theta) \,|\, \theta \in Crit(L) \cap \Omega\}.$$

## 4 CONCLUSIONS

We have introduced the notions of pure and spurious critical points as general tools for a geometric investigation of the landscape of neural networks. In particular, they provide a basic language for describing the interplay between a *convex loss function* and an *overparameterized, non-convex functional space*. In this paper, we have focused on the landscape of linear networks. This simple model is useful for illustrating our geometric perspective, but also exhibits several interesting (and surprisingly subtle) features. For example, the absence of non-global minima in the loss landscape is a rather general property when the architecture is "filling", while in the "non-filling" setting it is a special property that holds for the quadratic loss. Furthermore, we have observed that even in this simple framework global minima can have (possibly exponentially) many disconnected components.

In the future, we hope to extend our analysis to different network models. For example, we can use our framework to study networks with polynomial activations (Kileel et al., 2019), which are a direct generalization of the linear model. We expect that an analysis of pure and spurious critical points in this context can be used to address a conjecture in Venturi et al. (2018) regarding the gap between "upper" and "lower" dimensions in functional space. A geometric investigation of networks with smooth non-polynomial activations is also possible; in that setting, the parameter space and the functional space are usually of the same dimension (*i.e.*, $d_\theta = \dim(\mathcal{M}_\Phi)$), however there is still an interesting stratification of singular loci, as explained for example in (Amari, 2016, Section 12.2.2). General "discriminant hypersurfaces" can also be used to describe qualitative changes in the landscape as the data distribution varies. Finally, extending our analysis to networks with ReLU activations will require some care because of the non-differentiable setting. On the other hand, it is clear that ReLU networks behave as linear networks when restricted to appropriate regions of input space: this suggests that our study of ranks of differentials may be a useful building block for pursuing in this important direction.

**Acknowledgements.** We thank James Mathews for many helpful discussions in the beginning of this project. We are gratuful to ICERM (NSF DMS-1439786 and the Simons Foundation grant 507536) for the hospitality during the academic year 2018/2019 where many ideas for this project were developed. MT and JB were partially supported by the Alfred P. Sloan Foundation, NSF RI-1816753, NSF CAREER CIF 1845360, and Samsung Electronics. KK was partially supported by the Knut and Alice Wallenbergs Foundation within their WASP AI/Math initiative.

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

## A  APPENDIX

### A.1  DETERMINANTAL VARIETIES

We present some additional properties of determinantal varieties. For proofs and more details, we refer the reader to Harris (1995). Given $r < \min(m, n)$, the $r$-th determinantal variety $\mathcal{M}_r \subset \mathbb{R}^{m \times n}$ is defined as the set of matrices with rank at most $r$:

$$\mathcal{M}_r = \{P \in \mathbb{R}^{m \times n} \mid \mathrm{rk}(P) \leq r\} \subset \mathbb{R}^{m \times n}.$$

As mentioned in the main part of the paper, $\mathcal{M}_r$ is an algebraic variety of dimension $r(m + n - r)$, that can be described as the zero-set of all $(r + 1) \times (r + 1)$ minors. For $r > 0$, the the singular locus of $\mathcal{M}_r$ is exactly $\mathcal{M}_{r-1} \subset \mathcal{M}_r$. Some of our proofs will rely on the following explicit characterization of tangent space of determinantal varieties: given a a matrix $P \in \mathbb{R}^{m \times n}$ of rank exactly $r$ (so $P$ is a smooth point on $\mathcal{M}_r$) we have that

$$T_P \mathcal{M}_r = \mathbb{R}^m \otimes Row(P) + Col(P) \otimes \mathbb{R}^n \subset \mathbb{R}^{m \times n}.$$

We will also make use of the normal space to the tangent space $T_P \mathcal{M}_r$ at $P$, with respect to the Frobenius inner product. This is given by

$$(T_P \mathcal{M}_r)^\perp = Col(P)^\perp \otimes Row(P)^\perp,$$

where $Col(P)^\perp$ and $Row(P)^\perp$ are the orthogonal spaces to $Col(P)$ and $Row(P)$, respectively.

### A.2  EUCLIDEAN DISTANCE DEGREES AND DISCRIMINANTS

In this section, we informally discuss some algebraic notions related to ED (Euclidean distance) degrees and discriminants. A detailed presentation can be found in Draisma et al. (2013). Given an algebraic variety $\mathcal{V} \subset \mathbb{R}^n$ and a point $u \in \mathbb{R}^n$, the number of real critical points of the distance function $h_u(p) = \|p - u\|^2$ restricted to $\mathcal{V}$ is only locally constant as $u$ varies. In general, the behavior changes when $u$ crosses the *caustic hypersurface*, or *ED (Euclidean distance) discriminant*, of $\mathcal{V}$. The ED discriminant can be defined over the complex numbers, and in this setting it is indeed always a hypersurface (*i.e.*, it has codimension one), however it can have higher codimension over the real numbers. For instance, for a circle in the complex plane with the origin as its center, a point $(u_1, u_2) \in \mathbb{C}^2$ is on the ED discriminant if and only if $u_1^2 + u_2^2 = 0$. This defines a curve in the complex plane whose real locus is a point (see right side of Figure 2). By the Eckart-Young Theorem (Theorem 12), the ED discriminant of the determinantal variety $\mathcal{M}_r$ is the locus of all matrices $Q_0$

with at least two coinciding singular values, so it is defined by the discriminant of $Q_0 Q_0^T$. As in the case of the circle, the ED discriminant of $\mathcal{M}_r$ has codimension two in $\mathbb{R}^{d_y \times d_x}$.

Over the complex numbers, the number of critical points of the distance function $h_u$ restricted to $\mathcal{V}$ is actually *the same* for every point $u \in \mathbb{C}^n$ not on the ED discriminant of $\mathcal{V}$. This quantity is known as the *ED degree* of the variety $\mathcal{V}$. For instance, a circle has ED degree two whereas an ellipse has ED degree four (on the left side of Figure 2, points $u$ outside of the caustic curve yield two real critical points and two imaginary critical points). The Eckart-Young Theorem (Theorem 12) tells us that the ED degree of the determinantal variety $\mathcal{M}_r \subset \mathbb{R}^{d_y \times d_x}$ is $\binom{m}{r}$ where $m = \min(d_x, d_y)$. As argued in the main part of the paper, this does not hold any longer after perturbing either the determinantal variety or the Euclidean distance slightly, even using only a linear change of coordinates. For an algebraic variety $\mathcal{V} \subset \mathbb{C}^n$, a linear change of coordinates is given by an automorphism $\varphi : \mathbb{C}^n \to \mathbb{C}^n$. For almost all such automorphisms (*i.e.*, for all $\varphi$ except those lying in some subvariety of $\mathrm{GL}(n, \mathbb{C})$) the ED degree of $\varphi(\mathcal{V})$ is *the same*; see Theorem 5.4 in Draisma et al. (2013). This quantity is known as the *general ED degree* of $\mathcal{V}$. For instance, almost all linear coordinate changes will deform a circle into an ellipse, such that the general ED degree of the circle is four.

In the above definition of the general ED degree, we fixed the standard Euclidean distance and perturbed the variety. Alternatively, we can fix the variety and change the standard Euclidean distance $\| \cdot \|$ to $\mathrm{dist}_\varphi = \|\varphi(\cdot)\|$. The new distance function $h_{\varphi,u}(p) = \mathrm{dist}_\varphi(p - u)^2$ from $u$ satisfies $h_{\varphi(u)}(\varphi(p)) = h_{\mathrm{id},\varphi(u)}(\varphi(p)) = h_{\varphi,u}(p)$. Hence, the ED degree of $\varphi(\mathcal{V})$ with respect to the standard Euclidean distance $\mathrm{dist}_{\mathrm{id}} = \| \cdot \|$ equals the ED degree of $\mathcal{V}$ with respect to the perturbed Euclidean distance $\mathrm{dist}_\varphi$. In particular, the general ED degree of $\mathcal{V}$ can be obtained by computing the ED degree after applying a sufficiently random linear change of coordinates on either the Euclidean distance or the variety $\mathcal{V}$ itself.

As in the case of a circle, the general ED degree of the determinantal variety $\mathcal{M}_r$ is *not* equal to the ED degree of $\mathcal{M}_r$. Furthermore, there is no known closed formula for the general ED degree of $\mathcal{M}_r$ only involving the parameters $d_x$, $d_y$ and $r$. In the special case of rank-one matrices, one can derive a closed expression from the Catanese-Trifogli formula (Theorem 7.8 in Draisma et al. (2013)): the general ED degree of $\mathcal{M}_1$ is

$$\sum_{s=0}^{d_x+d_y} (-1)^s (2^{d_x+d_y+1-s} - 1)(d_x + d_y - s)! \left[ \sum_{\substack{i+j=s \\ i \le d_x, \, j \le d_y}} \frac{\binom{d_x+1}{i}\binom{d_y+1}{j}}{(d_x - i)!(d_y - j)!} \right].$$

This expression yields 39 for $d_x = d_y = 3$, as mentioned in Example 13. For general $r$, formulas for the general ED degree of $\mathcal{M}_r$ involving Chern and polar classes can be found in Ottaviani et al. (2013); Draisma et al. (2013). A short algorithm to compute the general ED degree of $\mathcal{M}_r$ is given in Example 7.11 of Draisma et al. (2013); it uses a package for advanced intersection theory in the algebro-geometric software `Macaulay2` (Grayson & Stillman, 2019).

This discussion shows that the Eckart-Young Theorem is indeed very special. The intrinsic reason for this is that the determinantal variety $\mathcal{M}_r$ intersects the "isotropic quadric" associated with the standard Euclidean distance (*i.e.*, zero locus of $X_{1,1}^2 + \ldots + X_{d_x,d_y}^2$ in $\mathbb{C}^{d_x \times d_y}$) in a particular way (*i.e.*, non-transversely). Performing a random linear change of coordinates on either $\mathcal{M}_r$ or the isotropic quadric makes the intersection transverse. So the ED degree after the linear change of coordinates is the general ED degree of $\mathcal{M}_r$, and the Eckart-Young Theorem does not apply.

In summary, we have observed that the degeneration from an ellipse to a circle is analogous to the degeneration from a determinantal variety with a perturbed Euclidean distance to the determinantal variety with the standard Euclidean distance: in both cases, the ED degree drops because the situation becomes degenerate. Moreover, the ED discriminant drops dimension, which causes the special phenomenon that the number of real critical points is almost everywhere the same.

**Experiment 1.** In general, it is very difficult to describe the open regions in $\mathbb{R}^n$ that are separated by the ED discriminant of a variety $\mathcal{V} \subset \mathbb{R}^n$. Finding the "typical" number of real critical points for the distance function $h_u$ restricted to $\mathcal{V}$, requires the computation of the volumes of these open regions. In the current state of the art in real algebraic geometry, this is only possible for very particular varieties $\mathcal{V}$. For these reasons, and to get more insights on the typical number of real critical points of determinantal varieties with a perturbed Euclidean distance, we performed computational experiments with `Macaulay2` (Grayson & Stillman, 2019) in the situation of Example 13. We fixed the

Table 2: Number of critical points (columns) and number of minima (rows) in our experiments

| | | #(critical points) | | | | | | |
| | | 1 | 3 | 5 | 7 | 9 | 11 | 13 |
| --- | --- | --- | --- | --- | --- | --- | --- | --- |
| | **1** | 0 | 476 | 120 | 1 | 0 | 0 | 0 |
| #(local | **2** | 0 | 0 | 805 | 190 | 10 | 0 | 0 |
| minima) | **3** | 0 | 0 | 0 | 228 | 116 | 21 | 0 |
| | **4** | 0 | 0 | 0 | 0 | 16 | 12 | 5 |

determinantal variety $\mathcal{M}_1 \subseteq \mathbb{R}^{3 \times 3}$ of rank-one $(3 \times 3)$-matrices. In each iteration of the experiment, we picked a random automorphism $\varphi : \mathbb{R}^{3 \times 3} \to \mathbb{R}^{3 \times 3}$ and a random matrix $Q_0 \in \mathbb{R}^{3 \times 3}$. We first verified that the number of complex critical points of the perturbed quadratic distance function $h_{\varphi, Q_0}$ restricted to $\mathcal{M}_1$ is the expected number 39. After that, we computed the number of real critical points and the number of local minima among them. Our results for 2000 iterations can be found in Table 2 and Figure 3. Although this is a very rudimentary experiment in an extremely simple setting, it provides clear evidence that the number of local minima of the perturbed distance function is generally not one.

**Implementation details:** We note that our computations of real critical points and local minima involved numerical methods and might thus be affected by numerical errors. In our implementation we used several basic tests to rule out numerically bad iterations, so that we can report our results with high confidence. The entries of the random matrix $Q_0$ are independently and uniformly chosen among the integers in $Z = \{-10, -9, \ldots, 9, 10\}$. The random automorphism $\varphi$ is given by a matrix in $Z^{9 \times 9}$ whose entries are also chosen independently and uniformly at random.

## A.3 PURE AND SPURIOUS CRITICAL POINTS IN PREDICTOR SPACE

We illustrate a variation of our functional setting where the notions of pure and spurious can also be naturally applied. We consider a training sample $x_1, \ldots, x_N \in \mathbb{R}^{d_x}, y_1, \ldots, y_N \in \mathbb{R}$ (for notational simplicity we use $d_y = 1$ but this is not necessary). We then write an empirical risk of the form

$$L(\theta) = g(\hat{Y}(\theta), Y),$$

where $\hat{Y}(\theta) = (\Phi(\theta, x_1), \ldots, \Phi(\theta, x_N)) \in \mathbb{R}^N, Y = (y_1, \ldots, y_N) \in \mathbb{R}^N$ and $g : \mathbb{R}^N \times \mathbb{R}^N \to \mathbb{R}$ is a convex function. As $\theta$ varies, $\hat{Y}(\theta)$ defines a "predictor manifold" $\mathcal{Y} \subset \mathbb{R}^N$, which depends only on the input data $x_1, \ldots, x_N$, but not on $\theta$. The function $L(\theta)$ can be naturally seen as a composition

$$\mathbb{R}^{d_\theta} \xrightarrow{\eta} \mathcal{Y} \xrightarrow{g} \mathbb{R},$$

where $\eta(\theta) = \hat{Y}(\theta) \in \mathcal{Y}$. We may now distinguish again between "pure" and "spurious" critical points for $L$. In an underparameterized regime $d_\theta < N$, or if the input data $x_1, \ldots, x_N$ is in some way special, then $\mathcal{Y} \subsetneq \mathbb{R}^N$ is a submanifold (with singularities), and critical points may arise because we are restricting $g$ to $\mathcal{Y}$ (pure), or because of the parameterization map $\eta$ (spurious). In a highly overparameterized regime $d_\theta \gg N$ (which is usually the case in practice), we expect $\mathcal{Y} = \mathbb{R}^N$. This can be viewed as analogous to the "filling" situation described for linear networks in this paper. In particular, all critical points that are not global minima for $L$ are necessarily spurious, since $g|_{\mathcal{Y}} = g$ is convex.

## A.4 PROOF OF THEOREM 4

We first show Lemma 3 with help of the following general observation:

**Proposition 15.** *Let* $V_1^+ \subseteq V_2^+ \subseteq \ldots \subseteq V_h^+$ *and* $V_1^- \supseteq V_2^- \supseteq \ldots \supseteq V_h^-$ *be vector spaces with dimensions* $r_i^+ := \dim(V_i^+)$ *and* $r_i^- := \dim(V_i^-)$ *for* $i = 1, \ldots, h$. *Then we have*

$$\dim\left((V_1^+ \otimes V_1^-) + (V_2^+ \otimes V_2^-) + \ldots + (V_h^+ \otimes V_h^-)\right) = \sum_{i=1}^{h} r_i^+ r_i^- - \sum_{i=1}^{h-1} r_i^+ r_{i+1}^-.$$

*Proof.* We prove this assertion by induction on $h$. The base case ($h = 1$) is clear: $\dim(V_1^+ \otimes V_1^-) = r_1^+ r_1^-$. For the induction step, we set $V := (V_1^+ \otimes V_1^-) + \ldots + (V_{h-1}^+ \otimes V_{h-1}^-)$. The key observation is that the inclusions $V_1^+ \subseteq V_2^+ \subseteq \ldots \subseteq V_h^+$ and $V_1^- \supseteq V_2^- \supseteq \ldots \supseteq V_h^-$ imply that $V \cap (V_h^+ \otimes V_h^-) = V_{h-1}^+ \otimes V_h^-$. Hence, applying the induction hypothesis to $V$, we derive

$$\dim\left(V + \left(V_h^+ \otimes V_h^-\right)\right) = \dim(V) + \dim(V_h^+ \otimes V_h^-) - \dim(V_{h-1}^+ \otimes V_h^-)$$

$$= \left(\sum_{i=1}^{h-1} r_i^+ r_i^- - \sum_{i=1}^{h-2} r_i^+ r_{i+1}^-\right) + r_h^+ r_h^- - r_{h-1}^+ r_h^-$$

$$= \sum_{i=1}^{h} r_i^+ r_i^- - \sum_{i=1}^{h-1} r_i^+ r_{i+1}^-. \qquad \square$$

**Lemma 3.** *The dimension of the image of the differential $d\mu_{\boldsymbol{d}}$ at $\theta = (W_h, \ldots, W_1)$ is given by*

$$\mathrm{rk}(d\mu_{\boldsymbol{d}}(\theta)) = \sum_{i=1}^{h} \mathrm{rk}(W_{>i}) \cdot \mathrm{rk}(W_{<i}) - \sum_{i=1}^{h-1} \mathrm{rk}(W_{>i}) \cdot \mathrm{rk}(W_{<i+1}),$$

*where we use the convention that $W_{<1} = I_{d_0}$, $W_{>h} = I_{d_h}$ are the identity matrices of size $d_0$, $d_h$.*

*Proof.* The image of the differential $d\mu_{\boldsymbol{d}}(\theta)$ is given in (5). Due to

$$Col(\overline{W}) \subseteq Col(W_{>1}) \subseteq \ldots \subseteq Col(W_{>h-1}) = Col(W_h),$$

$$Row(\overline{W}) \subseteq Row(W_{<h}) \subseteq \ldots \subseteq Row(W_{<2}) = Row(W_1),$$

(7)

we can apply Proposition 15, which concludes the proof. $\qquad \square$

Now we provide a proof for Theorem 4, starting from a refinement of the last statement.

**Proposition 16.** *Let $r = \min\{d_i\}$, $\theta = (W_h, \ldots, W_1)$, $\overline{W} = \mu_{\boldsymbol{d}}(\theta)$, and $e = \mathrm{rk}\overline{W}$. The image of the differential $d\mu_{\boldsymbol{d}}$ at $\theta$ contains the tangent space $T_{\overline{W}}\mathcal{M}_e$ of $\mathcal{M}_e$ at $\overline{W}$. Furthermore, for every $\overline{W} \in \mathcal{M}_e \setminus \mathcal{M}_{e-1}$ there exists $\theta'$ such that $\mu_{\boldsymbol{d}}(\theta') = \overline{W}$ and the image of $d\mu_{\boldsymbol{d}}(\theta')$ is exactly $T_{\overline{W}}\mathcal{M}_e$.*

*Proof.* Due to (7) the image (5) of $d\mu_{\boldsymbol{d}}(\theta)$ always contains $\mathbb{R}^{d_h} \otimes Row(\overline{W}) + Col(\overline{W}) \otimes \mathbb{R}^{d_0} = T_{\overline{W}}\mathcal{M}_e$. Furthermore, there always exists $(W_h, \ldots, W_1) \in \mu_{\boldsymbol{d}}^{-1}(\overline{W})$ such that each $W_i$ has rank exactly $r$ and the containments in (7) are all equalities. For example, one way to achieve this is to consider any decomposition $\overline{W} = UV^T$ where $U \in \mathbb{R}^{d_h \times r}$ and $V = \mathbb{R}^{d_0 \times r}$ and then set $W_1 = [V \,|\, 0]^T$, $W_h = [U \,|\, 0]$, and $W_i = \begin{bmatrix} I_r & 0 \\ 0 & 0 \end{bmatrix}$ for $2 \leq i \leq h-1$, where $I_r$ is the $(r \times r)$-identity matrix and the zeros fill in the dimensions $(d_i \times d_{i-1})$ of $W_i$. $\qquad \square$

The next two propositions discuss the first part of Theorem 4, which distinguishes between the filling and the non-filling case.

**Proposition 17.** *Let $r = \min\{d_i\}$ and $\theta = (W_h, \ldots, W_1)$. In the non-filling case (i.e., if $r < \min\{d_h, d_0\}$) we have that $\mathrm{rk}(d\mu_{\boldsymbol{d}}(\theta)) < \dim\mathcal{M}_r$ if and only if $\mathrm{rk}(\mu_{\boldsymbol{d}}(\theta)) < r$.*

*Proof.* If $\mathrm{rk}(\mu_{\boldsymbol{d}}(\theta)) = r$, then Proposition 16 implies that the image of the differential $d\mu_{\boldsymbol{d}}(\theta)$ is the whole tangent space of $\mathcal{M}_r$ at $\mu_{\boldsymbol{d}}(\theta)$. To prove the other direction of the assertion, we assume that $\mathrm{rk}(\mu_{\boldsymbol{d}}(\theta)) < r$. Since $r < \min\{d_h, d_0\}$, there is some $i \in \{1, \ldots, h-1\}$ such that $d_i = r$. We view $\mu_{\boldsymbol{d}}$ as the following concatenation of the matrix multiplication maps:

$$\mathbb{R}^{d_h \times d_{h-1}} \times \ldots \times \mathbb{R}^{d_1 \times d_0} \xrightarrow{\mu_{i,1}} \mathbb{R}^{d_h \times d_i} \times \mathbb{R}^{d_i \times d_0} \xrightarrow{\mu_{i,2}} \mathbb{R}^{d_h \times d_0}, \tag{8}$$

where $\mu_{i,1} = \mu_{(d_h, \ldots, d_i)} \times \mu_{(d_i, \ldots, d_0)}$ and $\mu_{i,2} = \mu_{(d_h, d_i, d_0)}$. Since $\mathrm{rk}(\mu_{\boldsymbol{d}}(\theta)) < r$, we have that $\mathrm{rk}(W_{>i}) < r$ or $\mathrm{rk}(W_{<i+1}) < r$. Without loss of generality, we may assume the latter. So applying Lemma 3 to $\mu_{i,2}$ and $\theta' := \mu_{i,1}(\theta)$ yields

$$\mathrm{rk}(d\mu_{\boldsymbol{d}}(\theta)) \leq \mathrm{rk}(d\mu_{i,2}(\theta')) = \mathrm{rk}(W_{<i+1})\,(d_h - \mathrm{rk}(W_{>i})) + \mathrm{rk}(W_{>i})d_0$$

$$< r\,(d_h - \mathrm{rk}(W_{>i})) + \mathrm{rk}(W_{>i})d_0 = \mathrm{rk}(W_{>i})\,(d_0 - r) + r\,d_h$$

$$\leq r\,(d_0 - r) + r\,d_h = \dim\left(\mathcal{M}_r\right). \qquad \square$$

**Proposition 18.** *Let $r = \min\{d_i\}$ and $\theta = (W_h, \ldots, W_1)$. In the filling case (i.e., if $r = \min\{d_h, d_0\}$) we have that $\mathrm{rk}(d\mu_{\boldsymbol{d}}(\theta)) < d_h d_0$ if and only if there is some $i \in \{1, \ldots, h-1\}$ with $\mathrm{rk}(W_{>i}) < d_h$ and $\mathrm{rk}(W_{<i+1}) < d_0$.*

*Proof.* Let us first assume that $\mathrm{rk}(W_{>i}) < d_h$ and $\mathrm{rk}(W_{<i+1}) < d_0$ for some $i \in \{1, \ldots, h-1\}$. We view $\mu_{\boldsymbol{d}}$ as the concatenation of the matrix multiplication maps in (8). Applying Lemma 3 to $\mu_{i,2}$ and $\theta' := \mu_{i,1}(\theta)$ yields

$$\mathrm{rk}(d\mu_{\boldsymbol{d}}(\theta)) \leq \mathrm{rk}(d\mu_{i,2}(\theta')) = \mathrm{rk}(W_{<i+1})\,(d_h - \mathrm{rk}(W_{>i})) + \mathrm{rk}(W_{>i})d_0$$
$$< d_0\,(d_h - \mathrm{rk}(W_{>i})) + \mathrm{rk}(W_{>i})d_0 = d_h d_0.$$

Secondly, we assume the contrary, *i.e.*, that every $i \in \{1, \ldots, h-1\}$ satisfies $\mathrm{rk}(W_{>i}) = d_h$ or $\mathrm{rk}(W_{<i+1}) = d_0$. We observe the following 2 key properties which hold for all $i \in \{1, \ldots, h-1\}$:

$$\begin{aligned} \mathrm{rk}(W_{>i}) = d_h &\quad \Rightarrow \quad \forall j \geq i : \mathrm{rk}(W_{>j}) = d_h, \\ \mathrm{rk}(W_{<i+1}) = d_0 &\quad \Rightarrow \quad \forall j \leq i : \mathrm{rk}(W_{<j+1}) = d_0. \end{aligned} \tag{9}$$

We consider the index set $\mathcal{I} := \{i \in \{1, \ldots, h-1\} \mid \mathrm{rk}(W_{<i+1}) = d_0\}$. If $\mathcal{I} = \emptyset$, our assumption implies that $\mathrm{rk}(W_{>i}) = d_h$ for every $i \in \{1, \ldots, h\}$. So due to Lemma 3 we have

$$\mathrm{rk}(d\mu_{\boldsymbol{d}}(\theta)) = d_h \sum_{i=1}^{h} \mathrm{rk}(W_{<i}) - d_h \sum_{i=1}^{h-1} \mathrm{rk}(W_{<i+1}) = d_h \mathrm{rk}(W_{<1}) = d_h d_0.$$

If $\mathcal{I} \neq \emptyset$, we define $k := \max \mathcal{I}$. So for every $i \in \{k+1, \ldots, h-1\}$ we have $\mathrm{rk}(W_{<i+1}) < d_0$, and thus $\mathrm{rk}(W_{>i}) = d_h$ by our assumption. Moreover, due to (9), every $j \in \{0, \ldots, k\}$ satisfies $\mathrm{rk}(W_{<j+1}) = d_0$. Hence, Lemma 3 yields

$$\mathrm{rk}(d\mu_{\boldsymbol{d}}(\theta)) = \sum_{j=1}^{k} \mathrm{rk}(W_{>j})d_0 + d_h d_0 + \sum_{i=k+2}^{h} d_h \mathrm{rk}(W_{<i})$$
$$- \sum_{j=1}^{k} \mathrm{rk}(W_{>j})d_0 - \sum_{i=k+1}^{h-1} d_h \mathrm{rk}(W_{<i+1}) = d_h d_0. \qquad \square$$

**Example 19.** According to Proposition 18, the differential of the matrix multiplication map is surjective whenever $\mathrm{rk}(\overline{W}) = r$, but also for certain $\theta$ when $\mathrm{rk}(\overline{W}) < r$. For example, let us consider the map $\mu_{(2,2,2)} : \mathbb{R}^{2\times 2} \times \mathbb{R}^{2\times 2} \to \mathbb{R}^{2\times 2}$ and the two factorizations $\theta = (\left[\begin{smallmatrix} 1 & 1 \\ 1 & 1 \end{smallmatrix}\right], \left[\begin{smallmatrix} 1 & 0 \\ 0 & 1 \end{smallmatrix}\right])$ and $\theta' = (\left[\begin{smallmatrix} 1 & 0 \\ 1 & 0 \end{smallmatrix}\right], \left[\begin{smallmatrix} 1 & 1 \\ 0 & 0 \end{smallmatrix}\right])$ of the rank-one matrix $\left[\begin{smallmatrix} 1 & 1 \\ 1 & 1 \end{smallmatrix}\right]$. According to Proposition 18, the differential $d\mu_{(2,2,2)}(\theta)$ has maximal rank 4. So it is surjective, whereas $d\mu_{(2,2,2)}(\theta')$ is not. In fact, by Lemma 3, we have $\mathrm{rk}(d\mu_{(2,2,2)}(\theta')) = 3$.

**Theorem 4.** *Let $r = \min\{d_i\}$, $\theta = (W_h, \ldots, W_1)$, and $\overline{W} = \mu_{\boldsymbol{d}}(\theta)$.*

- *(Filling case) If $r = \min\{d_h, d_0\}$, the differential $d\mu_{\boldsymbol{d}}(\theta)$ has maximal rank equal to $\dim M_r = d_h d_0$ if and only if, for every $i \in \{1, 2, \ldots, h-1\}$, either $\mathrm{rk}(W_{>i}) = d_h$ or $\mathrm{rk}(W_{<i+1}) = d_0$ holds.*

- *(Non-filling case) If $r < \min\{d_h, d_0\}$, the differential $d\mu_{\boldsymbol{d}}(\theta)$ has maximal rank equal to $\dim M_r = r(d_h + d_0 - r)$ if and only if $\mathrm{rk}(\overline{W}) = r$.*

*Furthermore, in both situations, if $\mathrm{rk}(\overline{W}) = e < r$, then the image of $d\mu_{\boldsymbol{d}}(\theta)$ always contains the tangent space $T_{\overline{W}}\mathcal{M}_e$ of $\mathcal{M}_e \subset \mathcal{M}_r$ at $\overline{W}$.*

*Proof.* This is an amalgamation of Propositions 16, 17 and 18. $\square$

## A.5 PROOF OF THEOREM 5

In the following we use the notation from Theorem 5:

**Theorem 5.** *Let $r = \min\{d_i\}$. If $\mathrm{rk}(\overline{W}) = r$, then the set of $\boldsymbol{d}$-factorizations $\mu_{\boldsymbol{d}}^{-1}(\overline{W})$ of $\overline{W}$ has exactly $2^b$ path-connected components, where $b = \#\{i \mid d_i = r, \ 0 < i < h\}$. If $\mathrm{rk}(\overline{W}) < r$, then $\mu_{\boldsymbol{d}}^{-1}(\overline{W})$ is always path-connected.*

We also write $\mathrm{GL}^+(r)$ for the set of matrices in $\mathrm{GL}(r)$ with positive determinant. Analogously, we set $\mathrm{GL}^-(r) := \{G \in \mathrm{GL}(r) \mid \det(G) < 0\}$.

We first prove Theorem 5 in the case that $b = 0$. To show that $\mu_{\boldsymbol{d}}^{-1}(\overline{W})$ is path-connected in this case, we show the following stronger assertion: given two matrices $W$ and $W'$ of arbitrary rank and factorizations $\theta \in \mu_{\boldsymbol{d}}^{-1}(W)$ and $\theta' \in \mu_{\boldsymbol{d}}^{-1}(W')$, each path in the codomain of $\mu_{\boldsymbol{d}}$ from $W$ to $W'$ can be lifted to a path in the domain of $\mu_{\boldsymbol{d}}$ from $\theta$ to $\theta'$.

**Proposition 20** (Path Lifting Property). *If $b = 0$, then for every $W, W' \in \mathbb{R}^{d_y \times d_x}$, every $\theta \in \mu_{\boldsymbol{d}}^{-1}(W)$, every $\theta' \in \mu_{\boldsymbol{d}}^{-1}(W')$ and every continuous function $f : [0, 1] \to \mathbb{R}^{d_y \times d_x}$ with $f(0) = W$ and $f(1) = W'$, there is a continuous function $F : [-1, 2] \to \mathbb{R}^{d_y \times d_{h-1}} \times \ldots \times \mathbb{R}^{d_1 \times d_x}$ such that $F(-1) = \theta$, $F(2) = \theta'$, $\mu_{\boldsymbol{d}}(F(t)) = W$ for every $t \in [-1, 0]$, $\mu_{\boldsymbol{d}}(F(t)) = W'$ for every $t \in [1, 2]$ and $\mu_{\boldsymbol{d}}(F(t)) = f(t)$ for every $t \in [0, 1]$.*

*Proof.* Without loss of generality, we may assume that $d_y \leq d_x$. Then the assumption $b = 0$ means that $d_i > d_y$ for all $i = 1, \ldots, h - 1$.

We prove the assertion by induction on $h$. For the induction beginning, we consider the cases $h = 1$ and $h = 2$. If $h = 1$, then $\mu_{\boldsymbol{d}}$ is the identity and Proposition 20 is trivial. For $h = 2$, we construct explicit lifts of the given paths. We first show that there is a path in $\mu_{\boldsymbol{d}}^{-1}(W)$ from $\theta = (W_2, W_1)$ to some $(B_2, B_1)$ such that $B_2$ has full rank.

*Claim 1.* Let $h = 2$, $W \in \mathbb{R}^{d_y \times d_x}$ and $(W_2, W_1) \in \mu_{\boldsymbol{d}}^{-1}(W)$. Then there is $(B_2, B_1) \in \mu_{\boldsymbol{d}}^{-1}(W)$ with $\mathrm{rk}(B_2) = d_y$ and a continuous function $g : [0, 1] \to \mu_{\boldsymbol{d}}^{-1}(W)$ such that $g(0) = (W_2, W_1)$ and $g(1) = (B_2, B_1)$.

*Proof.* If $\mathrm{rk}(W_2) = d_y$, we have nothing to show. So we assume that $s := \mathrm{rk}(W_2) < d_y$. Without loss of generality, we may further assume that the first $s$ rows of $W_2$ have rank $s$. As $s < d_1$, we find a matrix $G \in \mathrm{GL}^+(d_1)$ such that $W_2 G = \left[\begin{smallmatrix} I_s & 0 \\ M & 0 \end{smallmatrix}\right]$, where $I_s \in \mathbb{R}^{s \times s}$ is the identity matrix and $M \in \mathbb{R}^{(d_y - s) \times s}$. Since $\mathrm{GL}^+(d_1)$ is path-connected, there is a continuous function $g_1' : [0, 1] \to \mathrm{GL}^+(d_1)$ with $g_1'(0) = I_{d_1}$ and $g_1'(1) = G$. Concatenation with $\mathrm{GL}(d_1) \to \mu_{\boldsymbol{d}}^{-1}(W)$, $H \mapsto (W_2 H, H^{-1} W_1)$ yields a continuous path $g_1$ in $\mu_{\boldsymbol{d}}^{-1}(W)$ from $(W_2, W_1)$ to $(W_2 G, G^{-1} W_1)$.

Since $(W_2 G)(G^{-1} W_1) = W$, we see that $G^{-1} W_1 = \left[\begin{smallmatrix} W_{(s)} \\ N \end{smallmatrix}\right]$, where $W_{(s)} \in \mathbb{R}^{s \times d_x}$ is the first $s$ rows of $W$ and $N \in \mathbb{R}^{(d_y - s) \times d_x}$. Replacing $N$ by an arbitrary matrix $N'$ still yields that $W_2 G \left[\begin{smallmatrix} W_{(s)} \\ N' \end{smallmatrix}\right] = W$. Hence, we find a continuous path $g_2$ in $\mu_{\boldsymbol{d}}^{-1}(W)$ from $(W_2 G, G^{-1} W_1)$ to $(W_2 G, B_1 := \left[\begin{smallmatrix} W_{(s)} \\ 0 \end{smallmatrix}\right])$.

Finally, we can replace the 0-columns in $W_2 G$ by arbitrary matrices $M_1 \in \mathbb{R}^{s \times (d_1 - s)}$ and $M_2 \in \mathbb{R}^{(d_y - s) \times (d_1 - s)}$ such that $\left[\begin{smallmatrix} I_s & M_1 \\ M & M_2 \end{smallmatrix}\right] B_1 = W$ still holds. In particular, we can pick $M_1 = 0$ and $M_2 = \left[\begin{smallmatrix} I_{d_y - s} & 0 \end{smallmatrix}\right]$ such that $B_2 := \left[\begin{smallmatrix} I_s & 0 \\ M & M_2 \end{smallmatrix}\right]$ has full rank $d_y$, and we find a continuous path $g_3$ in $\mu_{\boldsymbol{d}}^{-1}(W)$ from $(W_2 G, B_1)$ to $(B_2, B_1)$. Putting $g_1$, $g_2$ and $g_3$ together yields a path $g$ as desired in Claim 1. $\diamondsuit$

As $B_2$ has full rank, we find a matrix $H \in \mathrm{GL}^+(d_1)$ such that $B_2 H = \left[\begin{smallmatrix} I_{d_y} & 0 \end{smallmatrix}\right]$. As in the proof of Claim 1, we construct a continuous path in $\mu_{\boldsymbol{d}}^{-1}(W)$ from $(B_2, B_1)$ to $(B_2 H, H^{-1} B_1)$. Since $H^{-1} B_1 = \left[\begin{smallmatrix} W \\ N \end{smallmatrix}\right]$ for some $N \in \mathbb{R}^{(d_1 - d_y) \times d_x}$, we also find a continuous path in $\mu_{\boldsymbol{d}}^{-1}(W)$ from $(B_2 H, H^{-1} B_1)$ to $(B_2 H, \left[\begin{smallmatrix} W \\ 0 \end{smallmatrix}\right])$. All in all, we have constructed a continuous path $F_1$ in $\mu_{\boldsymbol{d}}^{-1}(W)$ from $(W_2, W_1)$ to $(\left[\begin{smallmatrix} I_{d_y} & 0 \end{smallmatrix}\right], \left[\begin{smallmatrix} W \\ 0 \end{smallmatrix}\right])$. Analogously, we find a continuous path $F_3$ in $\mu_{\boldsymbol{d}}^{-1}(W')$ between $(W_2', W_1')$ and $(\left[\begin{smallmatrix} I_{d_y} & 0 \end{smallmatrix}\right], \left[\begin{smallmatrix} W' \\ 0 \end{smallmatrix}\right])$. Finally, we define $F_2 : [0, 1] \to \mathbb{R}^{d_y \times d_1} \times \mathbb{R}^{d_1 \times d_x}$, $t \mapsto (\left[\begin{smallmatrix} I_{d_y} & 0 \end{smallmatrix}\right], \left[\begin{smallmatrix} f(t) \\ 0 \end{smallmatrix}\right])$ such that putting $F_1$, $F_2$ and $F_3$ together yields a path $F$ as desired in Proposition 20.

For the induction step, we view $\mu_{\boldsymbol{d}}$ as the concatenation of the following two matrix multiplication maps:

$$\mathbb{R}^{d_y \times d_{h-1}} \times \ldots \times \mathbb{R}^{d_1 \times d_x} \xrightarrow{\mu_{(d_h,\ldots,d_1)} \times \text{id}} \mathbb{R}^{d_y \times d_1} \times \mathbb{R}^{d_1 \times d_x} \xrightarrow{\mu_{(d_y,d_1,d_x)}} \mathbb{R}^{d_y \times d_x}.$$

We consider $\theta = (W_h, \ldots, W_1)$ and $\theta' = (W'_h, \ldots, W'_1)$, as well as $W_{>1} = W_h \cdots W_2$ and $W'_{>1} = W'_h \cdots W'_2$. Given a path $f : [0,1] \to \mathbb{R}^{d_y \times d_x}$ from $W = W_{>1}W_1$ to $W' = W'_{>1}W'_1$, we apply the induction beginning ($h = 2$) to $\mu_{(d_y,d_1,d_x)}$ to get a path $F_2 : [-0.5, 1.5] \to \mathbb{R}^{d_y \times d_1} \times \mathbb{R}^{d_1 \times d_x}$ such that $F_2(-0.5) = (W_{>1}, W_1)$, $F_2(1.5) = (W'_{>1}, W'_1)$, $\mu_{(d_y,d_1,d_x)}(F_2(t)) = W$ for all $t \in [-0.5, 0]$, $\mu_{(d_y,d_1,d_x)}(F_2(t)) = W'$ for all $t \in [1, 1.5]$ and $\mu_{(d_y,d_1,d_x)}(F_2(t)) = f(t)$ for all $t \in [0,1]$. Now we apply the induction hypothesis on $\mu_{(d_h,\ldots,d_1)}$ and the path from $W_{>1}$ to $W'_{>1}$ given by the first factor of $F_2$. This yields a path $F_1 : [-1, 2] \to \mathbb{R}^{d_y \times d_{h-1}} \times \ldots \times \mathbb{R}^{d_2 \times d_1}$ with $F_1(-1) = (W_h, \ldots W_2)$, $F_1(2) = (W'_h, \ldots W'_2)$, $\mu_{(d_h,\ldots,d_1)}(F_1(t)) = W_{>1}$ for all $t \in [-1, -0.5]$, $\mu_{(d_h,\ldots,d_1)}(F_1(t)) = W'_{>1}$ for all $t \in [1.5, 2]$ and $\mu_{(d_h,\ldots,d_1)}(F_1(t))$ is the first factor of $F_2(t)$ for all $t \in [-0.5, 1.5]$. This allows us to define a continuous path $F : [-1, 2] \to \mathbb{R}^{d_y \times d_{h-1}} \times \ldots \times \mathbb{R}^{d_1 \times d_x}$ from $\theta$ to $\theta'$ by setting $F(t) = (F_1(t), W_1)$ for all $t \in [-1, -0.5]$, $F(t) = (F_1(t), W'_1)$ for all $t \in [1.5, 2]$ and for all $t \in [-1, -0.5]$ we let $F(t)$ consist of $F_1(t)$ and the second factor of $F_2(t)$. □

**Corollary 21.** *If $b = 0$, then $\mu_{\boldsymbol{d}}^{-1}(\overline{W})$ is path-connected for each $\overline{W} \in \mathbb{R}^{d_y \times d_x}$.*

*Proof.* Apply Proposition 20 to the constant function $f : [0,1] \to \mathbb{R}^{d_y \times d_x}, t \mapsto \overline{W}$. □

Now we study the case $b > 0$. We write $0 < i_1 < \ldots < i_b < h$ for those indices $i_j$ such that $d_{i_j} = r$. Then we view $\mu_{\boldsymbol{d}}$ as the concatenation of the following two matrix multiplication maps:

$$\mathbb{R}^{d_y \times d_{h-1}} \times \ldots \times \mathbb{R}^{d_1 \times d_x} \xrightarrow{\mu_1} \mathbb{R}^{d_y \times d_{i_b}} \times \mathbb{R}^{d_{i_b} \times d_{i_{b-1}}} \times \ldots \times \mathbb{R}^{d_{i_1} \times d_x} \xrightarrow{\mu_2} \mathbb{R}^{d_y \times d_x}, \qquad (10)$$

where $\mu_1 = \mu_{(d_h,\ldots,d_{i_b})} \times \mu_{(d_{i_b},\ldots,d_{i_{b-1}})} \times \ldots \times \mu_{(d_{i_1},\ldots,d_0)}$ and $\mu_2 = \mu_{(d_y,d_{i_b},\ldots,d_{i_1},d_x)}$. Applying the path lifting property described above to the map $\mu_1$, we will show in Proposition 26 that $\mu_2^{-1}(\overline{W})$ and $\mu_{\boldsymbol{d}}^{-1}(\overline{W})$ have the same number of (path-)connected components. So it remains to study the connected components of $\mu_2^{-1}(\overline{W})$. We can shortly write the map $\mu_2$ as

$$\mu_2 : \mathbb{R}^{d_y \times r} \times \left(\mathbb{R}^{r \times r}\right)^{b-1} \times \mathbb{R}^{r \times d_x} \longrightarrow \mathbb{R}^{d_y \times d_x}.$$

In the case that $\text{rk}(\overline{W}) = r$, we use the following natural action of $\text{GL}(r)^b$ on $\mu_2^{-1}(\overline{W})$:

$$\text{GL}(r)^b \times \mu_2^{-1}(\overline{W}) \longrightarrow \mu_2^{-1}(\overline{W}),$$
$$((G_b, \ldots, G_1), (A_{b+1}, \ldots, A_1)) \longmapsto \left(A_{b+1}G_b, G_b^{-1}A_bG_{b-1}, \ldots, G_1^{-1}A_1\right). \qquad (11)$$

In fact, we show now that $\mu_2^{-1}(\overline{W})$ is the orbit of any element under this action if $\text{rk}(\overline{W}) = r$. From this we will deduce in Corollaries 23 and 24 that $\mu_2^{-1}(\overline{W})$ is homeomorphic to $\text{GL}(r)^b$ and thus has $2^b$ (path-)connected components if the matrix $\overline{W}$ has maximal rank $r$.

**Proposition 22.** *Let $b > 0$ and $\theta = (A_{b+1}, \ldots, A_1) \in \mathbb{R}^{d_y \times r} \times \left(\mathbb{R}^{r \times r}\right)^{b-1} \times \mathbb{R}^{r \times d_x}$ such that $\overline{W} = \mu_2(\theta)$ has maximal rank $r$. Then $\mu_2^{-1}(\overline{W})$ is the orbit of $\theta$ under the action defined in (11), i.e.,*
$$\mu_2^{-1}(\overline{W}) = \left\{ \left(A_{b+1}G_b, G_b^{-1}A_bG_{b-1}, \ldots, G_1^{-1}A_1\right) \mid G_1, \ldots, G_b \in \text{GL}(r) \right\}.$$

*Proof.* One inclusion, namely "$\supseteq$", is trivial. We prove the other inclusion "$\subseteq$" by induction on $b$. For the induction beginning ($b = 1$), we write $\overline{W} = \left[\begin{smallmatrix} W_{11} & W_{12} \\ W_{21} & W_{22} \end{smallmatrix}\right]$ where $W_{11} \in \mathbb{R}^{r \times r}$. Without loss of generality, we may assume that $\text{rk}(W_{11}) = r$. Similarly, we write $A_2 = \left[\begin{smallmatrix} A_{21} \\ A_{22} \end{smallmatrix}\right]$ and $A_1 = \left[\begin{smallmatrix} A_{11} & A_{12} \end{smallmatrix}\right]$ where $A_{i1} \in \mathbb{R}^{r \times r}$ for $i = 1, 2$. For $(A'_2, A'_1) \in \mu_2^{-1}(\overline{W})$, we write analogously $A'_2 = \left[\begin{smallmatrix} A'_{21} \\ A'_{22} \end{smallmatrix}\right]$ and $A'_1 = \left[\begin{smallmatrix} A'_{11} & A'_{12} \end{smallmatrix}\right]$. Due to $\text{rk}(W_{11}) = r$, we have that $\text{rk}(A_{21}) = r = \text{rk}(A'_{21})$. Hence, there is a matrix $G \in \text{GL}(r)$ such that $A'_{21} = A_{21}G$. This implies that $A_{21}GA'_{11} = W_{11} = A_{21}A_{11}$, so $A'_{11} = G^{-1}A_{11}$. Due to $A_{21}A_{12} = W_{12} = A_{21}GA'_{12}$, we get that $A'_{12} = G^{-1}A_{12}$. Finally,

$\mathrm{rk}(W_{11}) = r$ implies that $\mathrm{rk}(A_{11}) = r$, so $A_{22}A_{11} = W_{21} = A'_{22}G^{-1}A_{11}$ shows $A'_{22} = A_{22}G$. Thus we have shown that $A'_2 = A_2G$ and $A'_1 = G^{-1}A_1$.

For the induction step ($b > 1$), we consider $(A'_{b+1}, \ldots, A'_1) \in \mu_2^{-1}(\overline{W})$ and $A_{>1} = A_{b+1} \cdots A_2, A'_{>1} = A'_{b+1} \cdots A'_2 \in \mathbb{R}^{d_y \times r}$. Now we can apply the induction beginning to find $G_1 \in \mathrm{GL}(r)$ such that $A'_{>1} = A_{>1}G_1$ and $A'_1 = G_1^{-1}A_1$. As $A'_{>1}$ has rank $r$ and $A'_{b+1} \cdots A'_2 = A'_{>1} = A_{b+1} \cdots (A_2G_1)$, we can apply the induction hypothesis on the map which multiplies the left-most $b$ matrices. This yields $G_b, \ldots G_2 \in \mathrm{GL}(r)$ such that $A'_{b+1} = A_{b+1}G_b, \ldots, A'_3 = G_3^{-1}A_3G_2, A'_2 = G_2^{-1}(A_2G_1)$. $\square$

**Corollary 23.** *If $b > 0$ and $\overline{W} \in \mathbb{R}^{d_y \times d_x}$ has maximal rank $r$, then $\mu_2^{-1}(\overline{W})$ is homeomorphic to $\mathrm{GL}(r)^b$.*

*Proof.* We fix $\theta = (A_{b+1}, \ldots, A_1) \in \mu_2^{-1}(\overline{W})$. The map $\varphi : \mathrm{GL}(r)^b \to \mu_2^{-1}(\overline{W})$ given by the action (11) on $\theta$ is continuous. We now construct its inverse. As $\mathrm{rk}(\overline{W}) = r$, we have that $\mathrm{rk}(A_i) = r$ for all $i = 1, \ldots, b+1$. Without loss of generality, we may assume that the first $r$ rows of $A_{b+1}$ have rank $r$. We write $\pi : \mathbb{R}^{d_y \times r} \to \mathbb{R}^{r \times r}$ for the projection which forgets the last $d_y - r$ rows of a given matrix. For $\theta' = (A'_{b+1}, \ldots, A'_1) \in \mu_2^{-1}(\overline{W})$, Proposition 22 shows that $\theta' = \varphi(G_b, \ldots, G_1)$ for some $(G_b, \ldots, G_1) \in \mathrm{GL}(r)^b$. So we have that $G_b = \pi(A_{b+1})^{-1}\pi(A'_{b+1}), G_{b-1} = A_b^{-1}G_bA'_b, \ldots, G_1 = A_2^{-1}G_2A'_2$, which defines a map

$$\psi : \mu_2^{-1}(\overline{W}) \longrightarrow \mathrm{GL}(r)^b,$$
$$(A'_{b+1}, \ldots, A'_1) \longmapsto \left(G(A'_{b+1}), \ A_b^{-1}G(A'_{b+1})A'_b, \ \ldots, \ A_2^{-1} \cdots A_b^{-1}G(A'_{b+1})A'_b \cdots A'_2\right),$$

where $G(A'_{b+1}) := \pi(A_{b+1})^{-1}\pi(A'_{b+1})$. By construction, $\psi$ is the inverse of $\varphi$. Since $\psi$ is continuous, it is a homeomorphism between $\mu_2^{-1}(\overline{W})$ and $\mathrm{GL}(r)^b$. $\square$

**Corollary 24.** *If $b > 0$ and $\overline{W} \in \mathbb{R}^{d_y \times d_x}$ has maximal rank $r$, then $\mu_2^{-1}(\overline{W})$ has $2^b$ connected components. Each of these components is path-connected.*

*Proof.* The group $\mathrm{GL}(r)$ has two connected components, namely $\mathrm{GL}^+(r)$ and $\mathrm{GL}^-(r)$. Both components are path-connected. Hence, $\mathrm{GL}(r)^b$ has $2^b$ connected components, each of them path-connected. By Corollary 23, the same holds for $\mu_2^{-1}(\overline{W})$. $\square$

To complete our understanding of the connected components of $\mu_2^{-1}(\overline{W})$, we consider the case that the matrix $W \in \mathbb{R}^{d_y \times d_x}$ does not have maximal rank $r$. In that case, it turns out that $\mu_2^{-1}(\overline{W})$ is path-connected, which we show by constructing explicit paths between any two elements of $\mu_2^{-1}(\overline{W})$.

**Proposition 25.** *Let $\overline{W} \in \mathbb{R}^{d_y \times d_x}$. If $b > 0$ and $\mathrm{rk}(\overline{W}) < r$, then $\mu_2^{-1}(\overline{W})$ is path-connected.*

*Proof.* We write $\overline{W} = \begin{bmatrix} \overline{W}_1 \\ \overline{W}_2 \end{bmatrix}$ where $\overline{W}_1 \in \mathbb{R}^{r \times d_x}$, and denote by $e$ the rank of $\overline{W}$. If $\mathrm{rk}(\overline{W}_1) = e$, then $\overline{W}_2 = M\overline{W}_1$ for some $M \in \mathbb{R}^{(d_y - r) \times r}$.

*Claim 2.* If $b = 1$, $(A_2, A_1) \in \mu_2^{-1}(\overline{W})$, $\mathrm{rk}(\overline{W}_1) = e$ and $\overline{W}_2 = M\overline{W}_1$, then there is a continuous function $f : [0, 1] \to \mu_2^{-1}(\overline{W})$ with $f(0) = (A_2, A_1)$ and $f(1) = (\begin{bmatrix} I_r \\ M \end{bmatrix}, \overline{W}_1)$.

*Proof.* Since $\mathrm{rk}(\overline{W}) < r$, we have that $\mathrm{rk}(A_2) < r$ or $\mathrm{rk}(A_1) < r$. If $\mathrm{rk}(A_2) < r$, we proceed as in the proof of Claim 1 to find a path in $\mu_2^{-1}(\overline{W})$ from $(A_2, A_1)$ to $(A'_2, A'_1)$ such that $\mathrm{rk}(A'_2) = r$.

Hence, we may assume that $\mathrm{rk}(A_2) = r$. This implies that $\mathrm{rk}(A_1) = e$. So $K := \ker(A_1^T) \subseteq \mathbb{R}^r$ has positive dimension $r - e$. We write $A_2 = \begin{bmatrix} A_{21} \\ A_{22} \end{bmatrix}$ where $A_{21} \in \mathbb{R}^{r \times r}$, and denote by $r_2$ the rank of $A_{21}$. So the rowspace $R \subseteq \mathbb{R}^r$ of $A_{21}$ has dimension $r_2$. We now show that $K + R = \mathbb{R}^r$. To see this we set $\delta := \dim(K \cap R)$. Without loss of generality, we may assume that the first $e$ rows of $\overline{W}_1$ are linearly independent. Then the first $e$ rows of $A_{21}$ must also be linearly independent, so we might further assume that the first $r_2$ rows of $A_{21}$ are linearly independent. We denote by $A_{211}$ and $\overline{W}_{11}$ the matrices formed by the first $r_2$ rows of $A_{21}$ and $\overline{W}_1$, respectively. In particular, we have

that $A_{211}A_1 = \overline{W}_{11}$. Now we choose a basis $(b_1, \ldots, b_{r_2})$ for $R$ such that $(b_1, \ldots, b_\delta)$ is a basis for $K \cap R$. Since $R$ is the rowspace of $A_{211}$, there is some $G \in \mathrm{GL}(r_2)$ such that the $i$-th row of $GA_{211}$ is $b_i$. So the first $\delta$ rows of $G\overline{W}_{11} = GA_{211}A_1$ are zero, which shows that $e = \mathrm{rk}(G\overline{W}_{11}) \le r_2 - \delta$. Thus, $\dim(K + R) = (r - e) + r_2 - \delta \ge r$, which proves that $K + R = \mathbb{R}^r$.

If $r_2 < r$, we now show that there is a path in $\mu_2^{-1}(\overline{W})$ from $(A_2, A_1)$ to $(A_2'' = \begin{bmatrix} A_{21}'' \\ A_{22} \end{bmatrix}, A_1)$ such that $\mathrm{rk}(A_{21}'') = r$. We may assume again that the first $r_2$ rows $a_1, \ldots, a_{r_2}$ of $A_{21}$ are linearly independent, *i.e.*, that they form a basis for $R$. Since $K + R = \mathbb{R}^r$, we can extend this basis to a basis $(a_1, \ldots, a_r)$ for $\mathbb{R}^r$ such that $a_i \in K$ for all $i > r_2$. We define $A_{21}''$ such that its first $r_2$ rows are $a_1, \ldots, a_{r_2}$ and such that its $i$-th row, for $r_2 < i \le r$, is the sum of $a_i \in K$ and the $i$-th row of $A_{21}$. Then $A_2'' = \begin{bmatrix} A_{21}'' \\ A_{22} \end{bmatrix}$ satisfies $A_2''A_1 = \overline{W}$. Moreover, the straight line from $(A_2, A_1)$ to $(A_2'', A_1)$ is a path in $\mu_2^{-1}(\overline{W})$. Since the last $r - r_2$ rows of $A_{21}$ are contained in the linear span $R$ of the first $r_2$ rows of $A_{21}$, the linearity of the determinant implies that $\det(A_{21}'') = \det([a_1 \cdots a_r]) \ne 0$.

Thus, we may assume that $r_2 = r$. If $\det(A_{21}) < 0$, we now construct a path in $\mu_2^{-1}(\overline{W})$ from $(A_2, A_1)$ to $(A_2''' = \begin{bmatrix} A_{21}''' \\ A_{22} \end{bmatrix}, A_1)$ such that $\det(A_{21}''') > 0$. For this, we pick a vector $v \in K \setminus \{0\}$. Since the rows of $A_{21}$ form a basis for $\mathbb{R}^r$, there is an index $i \in \{1, \ldots, r\}$ such that the matrix $D \in \mathbb{R}^{r \times r}$ obtained from $A_{21}$ by replacing its $i$-th row with $v$ has full rank. We pick $\mu \in \mathbb{R}$ such that $\det(A_{21}) + \mu \det(D) > 0$ and define $A_{21}'''$ to be the matrix obtained from $A_{21}$ by adding $\mu v$ to its $i$-th row. Then $\det(A_{21}''') = \det(A_{21}) + \mu \det(D) > 0$ and $A_2''' = \begin{bmatrix} A_{21}''' \\ A_{22} \end{bmatrix}$ satisfies $A_2'''A_1 = \overline{W}$. Moreover, the straight line from $(A_2, A_1)$ to $(A_2''', A_1)$ is a path in $\mu_2^{-1}(\overline{W})$.

Therefore, we may assume that $\det(A_{21}) > 0$, so $G := A_{21}^{-1} \in \mathrm{GL}^+(r)$. Any path in $\mathrm{GL}^+(r)$ from $I_r$ to $G$ yields a path in $\mu_2^{-1}(\overline{W})$ from $(A_2, A_1)$ to $(A_2 G, G^{-1}A_1) = (\begin{bmatrix} I_r \\ A_{22}G \end{bmatrix}, \overline{W}_1)$. Since $(A_{22}G)\overline{W}_1 = \overline{W}_2 = M\overline{W}_1$, the straight line from $(\begin{bmatrix} I_r \\ A_{22}G \end{bmatrix}, \overline{W}_1)$ to $(\begin{bmatrix} I_r \\ M \end{bmatrix}, \overline{W}_1)$ is a path in $\mu_2^{-1}(\overline{W})$. $\diamondsuit$

*Claim 3.* If $\theta = (A_{b+1}, \ldots, A_1) \in \mu_2^{-1}(\overline{W})$, $\mathrm{rk}(\overline{W}_1) = e$ and $\overline{W}_2 = M\overline{W}_1$, then there is a continuous function $F : [0, 1] \to \mu_2^{-1}(\overline{W})$ with $F(0) = \theta$ and $F(1) = (\begin{bmatrix} I_r \\ M \end{bmatrix}, I_r, \ldots, I_r, \overline{W}_1)$.

*Proof.* As $e < r$, at least one of the $A_i$ has rank smaller than $r$. We set $k := \min\{i \in \{1, \ldots, b+1\} \mid \mathrm{rk}(A_i) < r\}$. If $k < b + 1$, we first show that there is a path in $\mu_2^{-1}(\overline{W})$ from $\theta$ to $(A_{b+1}', \ldots, A_1')$ such that $\min\{i \in \{1, \ldots, b+1\} \mid \mathrm{rk}(A_i') < r\} = b + 1$. Since $\mathrm{rk}(A_k) < r$, the rank of $\overline{W}' := A_{k+1}A_k$ is smaller than $r$. We write $\overline{W}' = [\overline{W}_1' \ \overline{W}_2']$ where $\overline{W}_1'$ has $r$ columns. Without loss of generality, we may assume that $\mathrm{rk}(\overline{W}_1') = \mathrm{rk}(\overline{W}')$. Then there is a matrix $N$ such that $\overline{W}_2' = \overline{W}_1'N$. Hence, we can apply the transposed version of Claim 2, which yields a path from $(A_{k+1}, A_k)$ to $(\overline{W}_1', [I_r \ N])$ in the set of factorizations of $\overline{W}'$. Defining $\tilde{A}_{k+1} := \overline{W}_1'$, $\tilde{A}_k := [I_r \ N]$ and $\tilde{A}_i := A_i$ for all $i \in \{1, \ldots, b+1\} \setminus \{k, k+1\}$, extends this path to a path in $\mu_2^{-1}(\overline{W})$ from $\theta$ to $(\tilde{A}_{b+1}, \ldots, \tilde{A}_1)$ such that $\min\{i \in \{1, \ldots, b+1\} \mid \mathrm{rk}(\tilde{A}_i) < r\} = k + 1$. We note that this construction increased the number $k$. So we can repeat the construction until we reach $(A_{b+1}', \ldots, A_1')$ as desired.

Hence, we may assume that $k = b + 1$. Since $\mathrm{rk}(A_{b+1}) < r$, the rank of $\overline{W}'' := A_{b+1}A_b$ is smaller than $r$. We write $\overline{W}'' = \begin{bmatrix} \overline{W}_1'' \\ \overline{W}_2'' \end{bmatrix}$ where $\overline{W}_1''$ has $r$ rows. Since $\overline{W}_1 = \overline{W}_1''A_{b-1}\cdots A_1$ and the matrices $A_{b-1}, \ldots, A_1$ have rank $r$, we have that $\mathrm{rk}(\overline{W}_1'') = \mathrm{rk}(\overline{W}_1) = e$. Analogously, $\mathrm{rk}(\overline{W}'') = e$. So there is matrix $M'$ such that $\overline{W}_2'' = M'\overline{W}_1''$. Applying Claim 2 yields a path from $(A_{b+1}, A_b)$ to $(\begin{bmatrix} I_r \\ M' \end{bmatrix}, \overline{W}_1'')$ in the set of factorizations of $\overline{W}''$. This path can be extended to a path in $\mu_2^{-1}(\overline{W})$ from $\theta$ to $\theta' := (\begin{bmatrix} I_r \\ M' \end{bmatrix}, \overline{W}_1'', A_{b-1}, \ldots, A_1)$. Applying the same construction on $\overline{W}''' := \overline{W}_1''A_{b-1}$ yields a path in $\mu_2^{-1}(\overline{W})$ from $\theta'$ to $\theta'' := (\begin{bmatrix} I_r \\ M' \end{bmatrix}, I_r, \overline{W}_1''A_{b-1}, \ldots, A_1)$. We repeat the contruction until $(\begin{bmatrix} I_r \\ M' \end{bmatrix}, I_r, \ldots, I_r, \overline{W}_1)$ is reached. Since $M'\overline{W}_1 = \overline{W}_2 = M\overline{W}_1$, the straight line from $(\begin{bmatrix} I_r \\ M' \end{bmatrix}, I_r, \ldots, I_r, \overline{W}_1)$ to $(\begin{bmatrix} I_r \\ M \end{bmatrix}, I_r, \ldots, I_r, \overline{W}_1)$ is a path in $\mu_2^{-1}(\overline{W})$. $\diamondsuit$

Now we finally show that $\mu_2^{-1}(\overline{W})$ is path-connected. Without loss of generality, we may assume that $\mathrm{rk}(\overline{W}_1) = e$, and we write $\overline{W}_2 = M\overline{W}_1$. For $\theta, \theta' \in \mu_2^{-1}(\overline{W})$, there are paths in $\mu_2^{-1}(\overline{W})$ from $\theta$ resp. $\theta'$ to $(\left[\begin{smallmatrix} I_r \\ M \end{smallmatrix}\right], I_r, \ldots, I_r, \overline{W}_1)$, so there is a path from $\theta$ to $\theta'$ in $\mu_2^{-1}(\overline{W})$. $\qquad\square$

To settle the proof of Theorem 5, it is left to show that $\mu_2^{-1}(\overline{W})$ and $\mu_{\boldsymbol{d}}^{-1}(\overline{W})$ have indeed the same number of (path-)connected components, as we promised earlier.

**Proposition 26.** *Let $b > 0$ and $\overline{W} \in \mathbb{R}^{d_y \times d_x}$. Then $\mu_{\boldsymbol{d}}^{-1}(\overline{W})$ and $\mu_2^{-1}(\overline{W})$ have the same number of connected components. Moreover, each of these components is path-connected.*

*Proof.* Let us denote the connected components of $\mu_2^{-1}(\overline{W})$ be $C_1, \ldots, C_k$. By Corollary 24 and Proposition 25, each of these components is path-connected. Using the notation in (10), we have that $\mu_{\boldsymbol{d}}^{-1}(\overline{W}) = \bigcup_{i=1}^{k} \mu_1^{-1}(C_i)$. Since the $\mu_1^{-1}(C_1), \ldots, \mu_1^{-1}(C_k)$ are pairwise disconnected, we see that $\mu_{\boldsymbol{d}}^{-1}(\overline{W})$ has at least $k$ disconnected components. It is left to show that each $\mu_1^{-1}(C_i)$ is path-connected. For this, let $\theta, \theta' \in \mu_1^{-1}(C_i)$ and $\sigma := \mu_1(\theta), \sigma' := \mu_1(\theta') \in C_i$. As $C_i$ is path-connected, there is a path in $C_i$ from $\sigma$ to $\sigma'$. The map $\mu_1$ is a direct product of $b + 1$ matrix multiplication maps. To each factor we can apply Proposition 20, which yields a path in $\mu_1^{-1}(C_i)$ from $\theta$ to $\theta'$. $\qquad\square$

**Corollary 27.** *Let $b > 0$ and $\overline{W} \in \mathbb{R}^{d_y \times d_x}$. If $\mathrm{rk}(\overline{W}) = r$, then $\mu_{\boldsymbol{d}}^{-1}(\overline{W})$ has $2^b$ connected components, and each of these components is path-connected. If $\mathrm{rk}(W) < r$, then $\mu_{\boldsymbol{d}}^{-1}(\overline{W})$ is path-connnected.*

*Proof.* Combine Corollary 24 and Propositions 25 and 26. $\qquad\square$

*Proof of Theorem 5.* Theorem 5 is an amalgamation of Corollaries 21 and 27. $\qquad\square$

A.6    PROOFS OF PROPOSITIONS 6, 7, 9, 10 AND 14

**Proposition 6.** *If $\theta$ is such that $d\mu_{\boldsymbol{d}}(\theta)$ has maximal rank (see Theorem 4), then $\theta \in Crit(L)$ if and only if $\overline{W} \in Crit(\ell|_{\mathcal{M}_r})$, and $\theta$ is a minimum (resp., saddle, maximum) for $L$ if and only if $\overline{W}$ is a minimum (resp., saddle, maximum) for $\ell|_{\mathcal{M}_r}$. If $\mathrm{rk}(\overline{W}) = r$ (which implies that $d\mu_{\boldsymbol{d}}(\theta)$ has maximal rank) and $\theta \in Crit(L)$, then all $\boldsymbol{d}$-factorizations of $\overline{W}$ also belong to $Crit(L)$.*

*Proof.* If $\mu_{\boldsymbol{d}}$ is a local submersion at $\theta$ onto $\mathcal{M}_r$, then there exists an open neighborhood $U$ of $\overline{W}$ in $\mathcal{M}_r$ and an open neighborhood $V$ of $\theta$ with the property that $\mu_{\boldsymbol{d}}$ acts as a projection from $V$ onto $U$ (see, *e.g.*, (Lee, 2003, Theorem 7.3)). From this, we easily deduce that $\theta$ is a minimum (resp. saddle, maximum) for $L$ if and only if $\overline{W} = \mu_{\boldsymbol{d}}(\theta)$ is a minimum (resp. saddle, maximum) for $\ell|_{\mathcal{M}_r}$. Finally, if $\mathrm{rk}(\overline{W}) = r$, then $d\mu_{\boldsymbol{d}}(\theta)$ has maximal rank for all $\theta \in \mu_{\boldsymbol{d}}^{-1}(\overline{W})$ by Theorem 4. $\qquad\square$

**Proposition 7.** *If $\theta \in Crit(L)$ with $\mathrm{rk}(\overline{W}) = e \leq r$, then $\overline{W} \in Crit(\ell|_{\mathcal{M}_e})$. In other words, if $\mathrm{rk}(\overline{W}) < r$, then $\theta \in Crit(L)$ implies that $\overline{W}$ is a critical point for the restriction of $\ell$ to a smaller determinantal variety $\mathcal{M}_e$ (which is in the singular locus of the functional space $\mathcal{M}_r$ in the non-filling case).*

*Proof.* According to Theorem 4, if $\mu_{\boldsymbol{d}}(\theta) = \overline{W}$ with $\mathrm{rk}(\overline{W}) = e$, then $Im(d\mu_{\boldsymbol{d}}(\theta)) \supset T_{\overline{W}}\mathcal{M}_e$. This means that $\theta \in Crit(L)$ implies that $\overline{W}$ is critical for $\ell|_{\mathcal{M}_e}$. $\qquad\square$

**Proposition 9.** *Let $\theta \in Crit(L)$ be such that $\mathrm{rk}(\overline{W}) < r$, and assume that $d\ell(\overline{W}) \neq 0$. Then, for any neighborhood $U$ of $\theta$, there exists $\theta'$ in $U$ such that $\mu_{\boldsymbol{d}}(\theta') = \overline{W}$ but $\theta' \notin Crit(L)$. In particular, $\theta$ is a saddle point.*

*Proof.* Our proof is a modification of an argument used in Zhang (2019). Let us first consider the case that $\mu_{\boldsymbol{d}}$ is filling, so $r = \min\{d_h, d_0\}$. Without loss of generality, we assume $r = d_0$. Recall that the image of $d\mu_{\boldsymbol{d}}(\theta)$ is given by

$$\mathbb{R}^{d_h} \otimes Row(W_{<h}) + \ldots + Col(W_{>i}) \otimes Row(W_{<i}) + \ldots + Col(W_{>1}) \otimes \mathbb{R}^{d_0}.$$

We first note that $Row(W_{<h}) \neq \mathbb{R}^{d_0}$, for otherwise $d\mu_{\boldsymbol{d}}(\theta)$ would be surjective, implying that $d\ell(\overline{W}) = 0$. We define $i = \max\{j \mid Row(W_{<j}) = \mathbb{R}^{d_0}\}$, so $1 \leq i < h$ (writing $W_{<1} = I_{d_0}$). We have that

$$d\ell(\overline{W})(Col(W_{>i}) \otimes Row(W_{<i})) = d\ell(\overline{W})(Col(W_{>i}) \otimes \mathbb{R}^{d_0}) = 0. \tag{12}$$

Since $Row(W_{<i+1}) \subsetneq \mathbb{R}^{d_0}$, we may find $w_i \in \mathbb{R}^{d_i}, w_i \neq 0$ such that $w_i^T W_i \ldots W_1 = 0$. We now fix $\epsilon > 0$ and $v_{i+1} \in \mathbb{R}^{d_{i+1}}$ arbitrarily, and define $\tilde{W}_{i+1} = W_{i+1} + \epsilon(v_{i+1} \otimes w_i)$. Clearly, we have that $\tilde{W}_{i+1} W_i \ldots W_1 = W_{i+1} W_i \ldots W_1$. If $(W_h, \ldots, \tilde{W}_{i+1}, W_i, \ldots, W_1)$ is also a critical point of $L$, then

$$d\ell(\overline{W})(Col(W_h, \ldots, \tilde{W}_{i+1}) \otimes \mathbb{R}^{d_0}) = 0. \tag{13}$$

Combining (12) and (13), we have that

$$d\ell(\overline{W})(Col(W_h, \ldots, W_{i+2}(v_{i+1} \otimes w_i)) \otimes \mathbb{R}^{d_0}) = 0.$$

If this were true for all $v_{i+1}$, then it would imply

$$d\ell(\overline{W})(Col(W_h, \ldots, W_{i+2}) \otimes \mathbb{R}^{d_0}) = 0. \tag{14}$$

Hence, we have either found an arbitrarily small perturbation $\theta'$ of $\theta$ as required in Proposition 9, or (14) must hold. In the latter case, we reapply the same argument for $\tilde{W}_{i+2} = W_{i+2} + \epsilon(v_{i+2} \otimes w_{i+1})$ where $w_{i+1} \neq 0$ and $w_{i+1}^T W_{i+1} \ldots W_1 = 0$. Again, we can either construct an arbitrarily small perturbation $\theta'$ of $\theta$ as required in Proposition 9, or we have $d\ell(\overline{W})(Col(W_{>i+2}) \otimes \mathbb{R}^{d_0}) = 0$. Proceeding this way we eventually arrive at $d\ell(\overline{W})(\mathbb{R}^{d_h} \otimes \mathbb{R}^{d_0}) = 0$ so $d\ell(\overline{W}) = 0$, which contradicts the hypothesis. Thus, at some point we must find an arbitrarily small perturbation $\theta'$ of $\theta$ as required in Proposition 9, which concludes the proof in the case that $\mu_{\boldsymbol{d}}$ is filling.

We now consider the case that $\mu_{\boldsymbol{d}}$ is not filling. We pick $i \in \{1, \ldots, h-1\}$ such that $d_i = r$, and write for simplicity $A = W_h \ldots W_{i+1}$ and $B = W_i \ldots W_1$. The assumption $\mathrm{rk}(\overline{W}) < r$ implies that $\mathrm{rk}(A) < r$ or $\mathrm{rk}(B) < r$, and we assume without loss of generality that $\mathrm{rk}(A) < r$. We define the map $L_B(W_h', \ldots, W_{i+1}') = \ell(W_h' \ldots W_{i+1}' B)$. We also introduce the map $\ell_B(A') = \ell(A'B)$ and the matrix multiplication map $\mu_{\boldsymbol{d}_A}$ where $\boldsymbol{d}_A = (d_h, \ldots, d_{i+1})$, so that $L_B = \ell_B \circ \mu_{\boldsymbol{d}_A}$. If $\theta$ is a critical point for $L$, then $\theta_A = (W_h, \ldots, W_{i+1})$ must be a critical point for $L_B$. We are thus in the position to apply the analysis carried out in the filling case. In particular, we have that either $\theta_A$ can be perturbed to $\tilde{\theta}_A$ such that $\mu_{\boldsymbol{d}_A}(\theta_A) = \mu_{\boldsymbol{d}_A}(\tilde{\theta}_A)$ but $\tilde{\theta}_A \notin Crit(L_B)$, or $d\ell_B(A) = 0$. In the former case, we have that $\theta' = (\tilde{\theta}_A, \theta_B)$ is not a critical point for $L$, and we are done. If instead $d\ell_B(A) = 0$, then we have that

$$d\ell(\overline{W})(\mathbb{R}^{d_h} \otimes Row(B)) = 0,$$

because the image of the differential of the map $A' \mapsto A'B$ is given by $\mathbb{R}^{d_h} \otimes Row(B)$. We now proceed in a similar manner as before. We have that $Col(W_{>i}) \subsetneq \mathbb{R}^{d_h}$, because we assumed that $W_{>i} = A$ had rank less than $r \leq d_h$. Thus, we may find $w_{i+1} \in \mathbb{R}^{d_i}$, $w_{i+1} \neq 0$ such that $W_h \ldots W_{i+1} w_{i+1} = 0$. We fix $\epsilon > 0$ and $v_i$ arbitrarily, and define $\tilde{W}_i = W_i + \epsilon(w_{i+1} \otimes v_i)$. We have that $W_h \ldots W_{i+1} \tilde{W}_i = W_h \ldots W_{i+1} W_i$. If for all choices of $v_i$ we have that $(W_h, \ldots, \tilde{W}_i, \ldots, W_1)$ is still a critical point for $L$, then we can deduce that

$$d\ell(\overline{W})(\mathbb{R}^{d_h} \otimes Row(W_{i-1} \ldots W_1)) = 0.$$

Repeating this reasoning, we obtain our result as before.

$\square$

**Proposition 10.** *Let $\ell$ be any smooth convex function, and let $L = \ell \circ \mu_{\boldsymbol{d}}$. If $\theta$ is a non-global local minimum for $L$, then necessarily $\mathrm{rk}(\overline{W}) = r$ (so $\theta$ is a* pure *critical point). In particular, $L$ has non-global minima if and only if $\ell|_{\mathcal{M}_r}$ has non-global minima.*

*Proof.* The first statement follows immediately from Proposition 9: if $\theta \in Crit(L)$ is a non-global local minimum, then necessarily $d\ell(\overline{W}) \neq 0$, and we conclude that $\mathrm{rk}(\overline{W}) = r$. For the second statement, we observe that if $\ell$ is a convex function, then $\theta$ is a local minimum for $L$ if and only if $\overline{W} = \mu_{\boldsymbol{d}}(\theta)$ is a local minimum for $\ell|_{\mathcal{M}_r}$. Indeed, if $\overline{W} = \mu_{\boldsymbol{d}}(\theta)$ is a local minimum for $\ell|_{\mathcal{M}_r}$, then it is always true that any $\theta \in \mu_{\boldsymbol{d}}^{-1}(\overline{W})$ is a local minimum. Conversely, if $\theta$ is a local minimum,

then from Proposition 9 we see that either $d\ell(\overline{W}) = 0$, in which case $\overline{W}$ is a (global) minimum because $\ell$ is convex, or $\overline{W}$ must have maximal rank. In the latter case, $d\mu_{\boldsymbol{d}}(\theta)$ would be surjective (by Theorem 4), so $\overline{W}$ would also be a local minimum for $\ell|_{\mathcal{M}_r}$ (see Proposition 6). Finally, it is clear that $\theta$ is also a global minimum for $L$ if and only if $\overline{W}$ is a global minimum for $\ell|_{\mathcal{M}_r}$.  □

**Proposition 14.** *If $L' = \ell \circ \nu_{\boldsymbol{d}}$ and $L = \ell \circ \mu_{\boldsymbol{d}}$, then the critical locus $Crit(L')$ is in "correspondence" with $Crit(L) \cap \Omega$, meaning that*

$$\{\nu_{\boldsymbol{d}}(\theta') \mid \theta' \in Crit(L')\} = \{\mu_{\boldsymbol{d}}(\theta) \mid \theta \in Crit(L) \cap \Omega\}.$$

*Proof.* Let us define

$$p : \Omega \to \mathbb{R}^{d_\theta}, (W_h, \ldots, W_1) \mapsto \left( W_h, \frac{W_{h-1}}{\|W_{h-1}\|}, \ldots, \frac{W_1}{\|W_1\|} \right),$$

$$q : \Omega \to \mathbb{R}^{d_\theta}, (W_h, \ldots, W_1) \mapsto \left( W_h \|W_{h-1}\| \cdots \|W_1\|, \frac{W_{h-1}}{\|W_{h-1}\|}, \ldots, \frac{W_1}{\|W_1\|} \right).$$

The image of both of these maps is $N = \{(W_h, \ldots, W_1) \mid \|W_i\| = 1, i = 1, \ldots, h-1\}$. In fact, both maps are submersions onto $N$. Since $\nu_{\boldsymbol{d}} = \mu_{\boldsymbol{d}} \circ p$ and $\mu_{\boldsymbol{d}} \circ q = \mu_{\boldsymbol{d}}|_\Omega = \nu_{\boldsymbol{d}} \circ q$, it is enough to show the following two assertions: 1) $\theta' \in Crit(L')$ if and only if $p(\theta') \in Crit(L)$; and 2) $\theta \in Crit(L) \cap \Omega$ if and only if $q(\theta) \in Crit(L')$.

For 1), we deduce from $L' = L \circ p$ that $dL'(\theta') = dL(p(\theta')) \circ dp(\theta') = 0$ if $dL(p(\theta')) = 0$, but this also holds conversely: if $dL'(\theta') = 0$, then $Im(dp(\theta'))$ is contained in $Ker(dL(p(\theta')))$. Since $q \circ p = p$ and both maps $p$ and $q$ are submersions, we have that $Im(dp(\theta')) = T_{p(\theta')}N = Im(dq(p(\theta')))$. Now it follows from $L \circ q = L|_\Omega$ that $dL(p(\theta')) = dL(p(\theta')) \circ dq(p(\theta')) = 0$. For 2), we can argue analogously, exchanging the roles of $L$ and $L'$ as well as $p$ and $q$.  □

### A.7 Proof of Theorem 12

We consider a fixed matrix $Q_0 \in \mathbb{R}^{d_y \times d_x}$ and a singular value decomposition (SVD) $Q_0 = U\Sigma V^T$. Here $U \in \mathbb{R}^{d_y \times d_y}$ and $V \in \mathbb{R}^{d_x \times d_x}$ are orthogonal and $\Sigma \in \mathbb{R}^{d_y \times d_x}$ is diagonal with decreasing diagonal entries $\sigma_1, \sigma_2, \ldots, \sigma_m$ where $m = \min(d_x, d_y)$. We also write shortly $[m] = \{1, 2, \ldots, m\}$ and denote by $[m]_r$ the set of all subsets of $[m]$ of cardinality $r$. For $\mathcal{I} \in [m]_r$, we define $\Sigma_{\mathcal{I}} \in \mathbb{R}^{d_y \times d_x}$ to be the diagonal matrix with entries $\sigma_{\mathcal{I},1}, \sigma_{\mathcal{I},2}, \ldots, \sigma_{\mathcal{I},m}$ where $\sigma_{\mathcal{I},i} = \sigma_i$ if $i \in \mathcal{I}$ and $\sigma_{\mathcal{I},i} = 0$ otherwise. These matrices yield the critical points of the function $h_{Q_0}(P) = \|P - Q_0\|^2$ restricted to the determinantal variety $\mathcal{M}_r$.

**Theorem 28.** *If $Q_0 \notin \mathcal{M}_r$, the critical points of $h_{Q_0}|_{\mathcal{M}_r}$ are all matrices of the form $U\Sigma_{\mathcal{I}}V^T$ where $Q_0 = U\Sigma V^T$ is a SVD and $\mathcal{I} \in [m]_r$. The local minima are the critical points with $\mathcal{I} = [r]$. They are all global minima.*

*Proof.* A matrix $P \in \mathcal{M}_r$ is a critical point if and only if $(Q_0 - P) \in T_P\mathcal{M}_r^\perp = Col(P)^\perp \otimes Row(P)^\perp$. If $P = \sum_{i=1}^r \sigma_i'(u_i' \otimes v_i')$ and $Q_0 - P = \sum_{j=1}^e \sigma_j''(u_j'' \otimes v_j'')$ are SVD decompositions with $\sigma_i' \neq 0$ and $\sigma_j'' \neq 0$, the column spaces of $P$ and $Q_0 - P$ are spanned by the $u_i'$ and $u_j''$, respectively. Similarly, the row spaces of $P$ and $Q_0 - P$ are spanned by the $v_i'$ and $v_j''$, respectively. So $P$ is a critical point if and only if the vectors $u_i', u_j''$ and $v_i', v_j''$ are orthonormal, *i.e.*, if

$$Q_0 = P + (Q_0 - P) = \sum_{i=1}^r \sigma_i'(u_i' \otimes v_i') + \sum_{j=1}^e \sigma_j''(u_j'' \otimes v_j'')$$

is a SVD of $Q_0$. This proves that the critical points are of the form $U\Sigma_{\mathcal{I}}V^T$ where $Q_0 = U\Sigma V^T$ is a SVD and $\mathcal{I} \in [m]_r$.

Since $h_{Q_0}(U\Sigma_{\mathcal{I}}V^T) = \|U\Sigma_{[m]\setminus\mathcal{I}}V^T\|^2 = \|\Sigma_{[m]\setminus\mathcal{I}}\|^2 = \sum_{i \notin \mathcal{I}} \sigma_i^2$, we see that the global minima are exactly the critical points selecting $r$ of the largest singular values of $Q_0$, *i.e.*, with $\mathcal{I} = [r]$. It is left to show that there are no other local minima. For this, we consider a critical point $P = U\Sigma_{\mathcal{I}}V^T$ such that at least one selected singular value $\sigma_i$ for $i \in \mathcal{I}$ is strictly smaller than $\sigma_r$. We will show now that $P$ cannot be a local minimum. Since $\sigma_i < \sigma_r$, there is some $j \in [r]$ such that $j \notin \mathcal{I}$. As

above, we write $u_k$ and $v_k$ for the columns of $U$ and $V^T$ such that $Q_0 = \sum_{k=1}^m \sigma_k(u_k \otimes v_k)$ and $P = \sum_{k \in \mathcal{I}} \sigma_k(u_k \otimes v_k)$. We consider rotations in the planes spanned by $u_i, u_j$ and $v_i, v_j$, respectively: for $a \in [0, \frac{\pi}{2}]$, we set $u^{(\alpha)} = \cos(\alpha)u_i + \sin(\alpha)u_j$ and $v^{(\alpha)} = \cos(\alpha)v_i + \sin(\alpha)v_j$. Note that $u^{(0)} = u_i$ and $u^{(\frac{\pi}{2})} = u_j$; analogously for $v^{(\alpha)}$. Next we define $\sigma^{(\alpha)} = \cos^2(\alpha)\sigma_i + \sin^2(\alpha)\sigma_j$ and

$$P_\alpha = \sum_{k \in \mathcal{I} \setminus \{i\}} \sigma_k(u_k \otimes v_k) + \sigma^{(\alpha)}\left(u^{(\alpha)} \otimes v^{(\alpha)}\right) \in \mathcal{M}_r.$$

We note that $P_0 = P$ and $P_{\frac{\pi}{2}} = U\Sigma_{\mathcal{I} \setminus \{i\} \cup \{j\}}V^T$ are both critical points of $h_{Q_0}|_{\mathcal{M}_r}$.

It remains to show that $h_{Q_0}(P_\alpha)$ as a function in $\alpha$ is strictly decreasing on the interval $[0, \frac{\pi}{2}]$. From

$$h_{Q_0}(P_\alpha) = \|\sum_{k \notin \mathcal{I}} \sigma_k(u_k \otimes v_k) + \sigma_i(u_i \otimes v_i) - \sigma^{(\alpha)}(u^{(\alpha)} \otimes v^{(\alpha)})\|^2$$

and $u^{(\alpha)} \otimes v^{(\alpha)} = \cos^2(\alpha)(u_i \otimes v_i) + \cos(\alpha)\sin(\alpha)(u_i \otimes v_j + u_j \otimes v_i) + \sin^2(\alpha)(u_j \otimes v_j)$, we deduce that

$$h_{Q_0}(P_\alpha) = \sum_{k \notin \mathcal{I}, k \neq j} \sigma_k^2 + \left(\sigma_i - \sigma^{(\alpha)}\cos^2(\alpha)\right)^2 + 2\left(\sigma^{(\alpha)}\cos(\alpha)\sin(\alpha)\right)^2 + \left(\sigma_j - \sigma^{(\alpha)}\sin^2(\alpha)\right)^2$$

$$= \sum_{k \notin \mathcal{I}, k \neq j} \sigma_k^2 + \sigma_i^2 + 2\sigma_j(\sigma_j - \sigma_i)\cos^2(\alpha) - (\sigma_j - \sigma_i)^2\cos^4(\alpha).$$

The graph of the function $f(x) = \sigma_i^2 + 2\sigma_j(\sigma_j - \sigma_i)x - (\sigma_j - \sigma_i)^2 x^2$, for $x \in \mathbb{R}$, is a parabola with a unique local and global maximum at $x_0 = \frac{\sigma_j}{\sigma_j - \sigma_i}$. Since $x_0 \geq 1$, the function $f$ is strictly increasing on the interval $[0, 1]$. Hence, $h_{Q_0}(P_\alpha) = \sum_{k \notin \mathcal{I}, k \neq j} \sigma_k^2 + f(\cos^2(\alpha))$ is strictly decreasing on $[0, \frac{\pi}{2}]$, which concludes the proof. $\qquad \square$

If the singular values of $Q_0$ are pairwise distinct and positive, the singular vectors of $Q_0$ are unique up to sign. So for each index set $\mathcal{I} \in [m]_r$ the matrix $Q_{\mathcal{I}} = U\Sigma_{\mathcal{I}}V^T$ is the unique critical point of $h_{Q_0}|_{\mathcal{M}_r}$ whose singular values are the $\sigma_i$ for $i \in \mathcal{I}$. Hence, Theorem 28 implies immediately the following:

**Corollary 29.** *If the singular values of $Q_0$ are pairwise distinct and positive, $h_{Q_0}|_{\mathcal{M}_r}$ has exactly $\binom{m}{r}$ critical points, namely the $Q_{\mathcal{I}} = U\Sigma_{\mathcal{I}}V^T$ for $\mathcal{I} \in [m]_r$. Moreover, its unique local and global minimum is $Q_{[r]}$.*

We can strengthen this result by explicitly calculating the *index* of each critical point, *i.e.*, the number of negative eigenvalues of the Hessian matrix.

**Theorem 30.** *If the singular values of $Q_0$ are pairwise distinct and positive, the index of $Q_{\mathcal{I}}$ as a critical point of $h_{Q_0}|_{\mathcal{M}_r}$ is*

$$\mathrm{index}(Q_{\mathcal{I}}) = \#\{(j, i) \in \mathcal{I} \times ([m] \setminus \mathcal{I}) \,|\, j > i\}.$$

To prove this assertion, we may assume without loss of generality that $d_y \leq d_x$, so $m = d_y$. We may further assume that $Q_0$ is a diagonal matrix, so $Q_0 = \Sigma$. Let $\mu_{(d_y, r, d_x)} : \mathbb{R}^{d_y \times r} \times \mathbb{R}^{r \times d_x} \to \mathbb{R}^{d_y \times d_x}$ be the matrix multiplication map, and $L = h_\Sigma \circ \mu_{(d_y, r, d_x)}$. For $(A, B) \in \mu_{(d_y, r, d_x)}^{-1}(\Sigma_{\mathcal{I}})$, Theorem 4 implies that the condition $\Sigma_{\mathcal{I}} \in Crit(h_\Sigma|_{\mathcal{M}_r})$ is equivalent to $dL(A, B) = 0$. Moreover, the number of negative eigenvalues of the Hessian of $L$ at any such factorization $(A, B)$ of $\Sigma_{\mathcal{I}}$ is the same. This number is the index of $\Sigma_{\mathcal{I}}$. So we can compute it by fixing one specific factorization $(A, B)$ of $\Sigma_{\mathcal{I}}$.

To compute the Hessian of $L$ at $(A, B)$, we compute the partial derivatives of first and second order of $L$:

$$\frac{\partial L}{\partial a_{ij}} = 2\left[(AB - \Sigma)B^T\right]_{ij}, \quad \frac{\partial L}{\partial b_{ij}} = 2\left[A^T(AB - \Sigma)\right]_{ij},$$

$$\frac{\partial^2 L}{\partial a_{ij} \partial a_{kl}} = \begin{cases} 0 & \text{, if } i \neq k \\ 2[BB^T]_{jl} & \text{, if } i = k \end{cases}, \tag{15}$$

$$\frac{\partial^2 L}{\partial b_{ij} \partial b_{kl}} = \begin{cases} 0 & \text{, if } j \neq l \\ 2[A^T A]_{ik} & \text{, if } j = l \end{cases}, \tag{16}$$

$$\frac{\partial^2 L}{\partial a_{ij} \partial b_{kl}} = \begin{cases} 2a_{ik}b_{jl} & \text{, if } j \neq k \\ 2\left(a_{ik}b_{jl} + [AB - \Sigma]_{il}\right) & \text{, if } j = k \end{cases}. \tag{17}$$

To assemble these second order partial derivatives into a matrix, we choose the following order of the entries of $(A, B)$:

$$a_{11}, \ldots, a_{1r}, \; a_{21}, \ldots, a_{2r}, \; \ldots, \; a_{d_y1}, \ldots, a_{d_yr}, \; b_{11}, \ldots, b_{r1}, \; b_{12}, \ldots, b_{r2}, \; \ldots, \; b_{1d_x}, \ldots, b_{rd_x}.$$

We denote by $H$ the Hessian matrix of $L$ with respect to this ordering at the following specifically chosen matrices $(A_0, B_0)$: denoting by $i_1, i_2, \ldots, i_r$ the entries of $\mathcal{I}$ in decreasing order, we pick the $j$-th column of $A_0$ to be the $i_j$-th standard basis vector in $\mathbb{R}^{d_y}$ and the $j$-th row of $B_0$ to be the $\sigma_{i_j}$-multiple of the $i_j$-th standard basis vector in $\mathbb{R}^{d_x}$. Note that $A_0 B_0 = \Sigma_{\mathcal{I}}$, $A_0^T A_0 = I_r$ is the $r \times r$-identity matrix, and $B_0 B_0^T$ is the $r \times r$-diagonal matrix with entries $\sigma_{i_1}^2, \sigma_{i_2}^2, \ldots, \sigma_{i_r}^2$. We write

$$H = \begin{bmatrix} D & M \\ M^T & N \end{bmatrix}, \quad \text{where } D \in \mathbb{R}^{rd_y \times rd_y}, N \in \mathbb{R}^{rd_x \times rd_x}, M \in \mathbb{R}^{rd_y \times rd_x}.$$

Our first observation is that $N$, whose entries are described by (16), is twice the identity matrix, so $N = 2I_{rd_x}$. Similarly, we see from (15) that $D$ is a diagonal matrix. According to our fixed ordering, the entries of $D$ are indexed by pairs $(ij, kl)$ of integers $i, k \in [d_y]$ and $j, l \in [r]$. With this, the diagonal entries of $D$ are $D_{ij,ij} = 2\sigma_{i_j}^2$. Analogously, the entries of $M$ are indexed by pairs $(ij, kl)$ of integers $i \in [d_y]$, $j, k \in [r]$ and $l \in [d_x]$.

**Lemma 31.** *Let $i \in [d_y]$ and $j \in [r]$. The $ij$-th row of $M$ has exactly one non-zero entry. If $i \in \mathcal{I}$, there is some $k \in [r]$ with $i = i_k$ and the non-zero entry is $M_{ij,ki_j} = 2\sigma_{i_j}$. Otherwise, so if $i \notin \mathcal{I}$, the non-zero entry is $M_{ij,ji} = -2\sigma_i$.*

*Proof.* The entries of $M$ are given by (17). We first observe that $(A_0)_{ik}(B_0)_{jl}$ is non-zero if and only if $i = i_k$ and $l = i_j$. Moreover, we have that $(A_0)_{i_k k}(B_0)_{ji_j} = \sigma_{i_j}$. Similarly, $[A_0 B_0 - \Sigma]_{il}$ is non-zero if and only $i = l \notin \mathcal{I}$. For $i \notin \mathcal{I}$ we have that $[A_0 B_0 - \Sigma]_{ii} = -\sigma_i$.

Now we fix $i$ and $j$ and consider the $ij$-th row of $M$. We apply our observations above to the following three cases.

If $i = i_j$, then $M_{ij,kl}$ is non-zero if and only if $k = j$ and $l = i$. In that case, $M_{ij,ji} = 2\sigma_i$.

If $i \in \mathcal{I}$, but $i \neq i_j$, then there is some $n \neq j$ such that $i = i_n$. Now $M_{ij,kl}$ is non-zero if and only if $k = n$ and $l = i_j$. In that case, $M_{ij,ki_j} = 2\sigma_{i_j}$.

Finally, if $i \notin \mathcal{I}$, then $M_{ij,kl}$ is non-zero if and only if $k = j$ and $l = i$. In that case, we have that $M_{ij,ji} = -2\sigma_i$. $\square$

**Corollary 32.** *The square matrix $\Delta := MM^T \in \mathbb{R}^{rd_y \times rd_y}$ is a diagonal matrix. For $i \in [d_y]$ and $j \in [r]$, its $ij$-th diagonal entry is $\Delta_{ij,ij} = 4\sigma_{i_j}^2$ if $i \in \mathcal{I}$ and $\Delta_{ij,ij} = 4\sigma_i^2$ if $i \notin \mathcal{I}$.*

*Proof.* The computation of the diagonal entries follows directly from Lemma 31. To see that all other entries of $\Delta$ are zero, we need to show that no column of $M$ has more than one non-zero entry. So let us assume for contradiction that the $kl$-th column of $M$ has non-zero entries in the $ij$-th row and in the $\bar{i}\bar{j}$-th row for $(i, j) \neq (\bar{i}, \bar{j})$.

If $i, \bar{i} \in \mathcal{I}$, then Lemma 31 implies $i = i_k = \bar{i}$ and $i_j = l = i_{\bar{j}}$, which contradicts $(i, j) \neq (\bar{i}, \bar{j})$.

If $i, \bar{i} \notin \mathcal{I}$, we see from Lemma 31 that $j = k = \bar{j}$ and $i = l = \bar{i}$, which contradicts $(i, j) \neq (\bar{i}, \bar{j})$.

Finally, if $i \in \mathcal{I}$ and $\bar{i} \notin \mathcal{I}$, then Lemma 31 yields that $\bar{i} = l = i_j \in \mathcal{I}$; a contradiction. $\square$

**Corollary 33.** *The characteristic polynomial of $H$ is*

$$(t - 2)^{r|d_x - d_y|} \cdot t^{r^2} \cdot \prod_{k \in \mathcal{I}} \left(t - 2(\sigma_k^2 + 1)\right)^r \cdot \prod_{i \in [m] \setminus \mathcal{I}} \prod_{j \in \mathcal{I}} \left(t^2 - 2t(\sigma_j^2 + 1) + 4(\sigma_j^2 - \sigma_i^2)\right). \tag{18}$$

*Proof.* Using Schur complements, we can compute the characteristic polynomial of $H$ as follows:

$$\chi_H(t) = \det\left(tI_{r(d_x+d_y)} - H\right)$$
$$= \det\left(tI_{rd_x} - 2I_{rd_x}\right)\det\left((tI_{rd_y} - D) - M(tI_{rd_x} - 2I_{rd_x})^{-1}M^T\right)$$
$$= (t-2)^{rd_x}\det\left((tI_{rd_y} - D) - (t-2)^{-1}\Delta\right)$$
$$= (t-2)^{r(d_x-d_y)}\det\left((t-2)(tI_{rd_y} - D) - \Delta\right).$$

By Corollary 32, the matrix $(t-2)(tI_{rd_y} - D) - \Delta$ is a diagonal matrix whose $ij$-th diagonal entry is $(t-2)(t - D_{ij,ij}) - \Delta_{ij,ij}$. We write shortly $\delta_{ij} := \Delta_{ij,ij}$ and use the identity $D_{ij,ij} = 2\sigma_{i_j}^2$ to further derive

$$\chi_H(t) = (t-2)^{r(d_x-d_y)}\prod_{i=1}^{d_y}\prod_{j=1}^{r}\left((t-2)(t - 2\sigma_{i_j}^2) - \delta_{ij}\right)$$

$$= (t-2)^{r(d_x-d_y)}\prod_{i=1}^{d_y}\prod_{j=1}^{r}\left(t^2 - 2t(\sigma_{i_j}^2 + 1) + (4\sigma_{i_j}^2 - \delta_{ij})\right)$$

$$= (t-2)^{r(d_x-d_y)}\left(\prod_{i\in\mathcal{I}}\prod_{j=1}^{r}\left(t\left(t - 2(\sigma_{i_j}^2 + 1)\right)\right)\right)$$

$$\cdot\left(\prod_{i\in[d_y]\setminus\mathcal{I}}\prod_{j=1}^{r}\left(t^2 - 2t(\sigma_{i_j}^2 + 1) + 4(\sigma_{i_j}^2 - \sigma_i^2)\right)\right).$$

The latter equality was derived by substituting specific values into the $\delta_{ij}$ according to Corollary 32. Rearranging the terms of this last expression of $\chi_H(t)$ yields (18). $\square$

**Lemma 34.** *Let $x, y > 0$. The polynomial $f(t) = t^2 - 2t(x+1) + 4(x-y)$ has two real roots and at least one of them is positive. Moreover, $f(t)$ has a negative root if and only if $x < y$.*

*Proof.* The roots of $f(t)$ are $x + 1 \pm \sqrt{(x+1)^2 - 4(x-y)} = x + 1 \pm \sqrt{(x-1)^2 + 4y}$. So the discriminant is positive and $f(t)$ has two real roots. Clearly, one of these is positive. The other one is negative if and only if $x + 1 < \sqrt{(x-1)^2 + 4y}$, which is equivalent to $(x+1)^2 < (x-1)^2 + 4y$ and thus to $x < y$. $\square$

*Proof of Theorem 30.* It is left to count the number of negative roots of the univariate polynomial (18). All the linear factors of (18) have non-negative roots. The $ij$-th quadratic factor of (18), for $i \in [d_y] \setminus \mathcal{I}$ and $j \in \mathcal{I}$, has at most one negative root due to Lemma 34. Moreover, it has exactly one negative root if and only if $\sigma_j^2 < \sigma_i^2$, which is equivalent to $j > i$. Hence, the polynomial (18) has exactly $\#\{(j,i) \in \mathcal{I} \times [d_y] \setminus \mathcal{I} \mid j > i\}$ many negative roots. $\square$

**Theorem 12.** *If the singular values of $Q_0$ are pairwise distinct and positive, $h_{Q_0}|_{\mathcal{M}_r}$ has exactly $\binom{m}{r}$ critical points, namely the matrices $Q_{\mathcal{I}} = U\Sigma_{\mathcal{I}}V^T$ with $\#(\mathcal{I}) = r$. Moreover, its unique local and global minimum is $Q_{\{1,\dots,r\}}$. More precisely, the index of $Q_{\mathcal{I}}$ as a critical point of $h_{Q_0}|_{\mathcal{M}_r}$ (i.e., the number of negative eigenvalues of the Hessian matrix for any local parameterization) is*

$$\mathrm{index}(Q_{\mathcal{I}}) = \#\{(j,i) \in \mathcal{I} \times \mathcal{I}^c \mid j > i\}, \qquad \text{where } \mathcal{I}^c = \{1,\dots,m\} \setminus \mathcal{I}.$$

*Proof.* This is an amalgamation of Corollary 29 and Theorem 30. $\square$

