# OpenReview forum: "Pure and Spurious Critical Points: a Geometric Study of Linear Networks"
_ICLR.cc/2020/Conference — Accept (Poster)_

### Official Review · AnonReviewer3 · 2019-10-22
**Official Blind Review #3**

**Rating:** 8

**Review:**

The authors use tools from algebraic geometry to study the critical points that arise in linear neural networks under a variety of loss functions.  In particular, they study linear maps that are parameterized (or overparameterized) as the product of multiple matrices.  The authors introduce a distinction between two types of critical points (for the linear function composed with a provided loss): there are those that are due to the actual manifold of functions that are provided to the loss (pure critical points), and there are those that are due to parameterizations of this manifold (spurious critical points).  The authors show that in the case that the linear function has maximal rank (is "filling"), then there are no non-global minima in the loss landscape.  In the case of lower rank ("nonfilling") linear functions, this property does not hold generally, but does hold for quadratic losses.   The authors show that for this framework, there can be exponentially many disconnected components in parameter space corresponding to global minima.

This paper provides important insight into the loss landscape of neural networks, and it exhibits nuanced observations even in the linear case.  Naturally, there are many important follow-up studies that can be done in the case of non-linear networks.  It is vital to have a thorough understanding before that work can be satisfactorily undertaken.

It would benefit the authors to provide more summary/commentary in the main results section.  Currently, that section comes across as a list of lemmas, theorems, propositions, and corollaries, with relatively little narration in between.  It was challenging for me to identify which of these items were the main results, as summarized on in the first paragraph of the conclusions section.  I recommend that any theorem/corollary/proposition that is the primary basis for a claim made in the conclusion should receive significantly more commentary in Section 3.

Minor comment:
In the first paragraph of the conclusion, it says that "... the absence of ... is a rather general property ...".  The word "rather" should be clarified.


**Experience Assessment:**

I do not know much about this area.

**Review Assessment: Checking Correctness Of Derivations And Theory:**

I did not assess the derivations or theory.

**Review Assessment: Checking Correctness Of Experiments:**

N/A

**Review Assessment: Thoroughness In Paper Reading:**

I made a quick assessment of this paper.

---

> ### Author Response · Authors · 2019-11-11
> **Author Response to Review #3**
>
> Thank you again for the comments, in addition to our general response given above, we reply here to some specific concerns:
>
> * "It would benefit the authors to provide more summary/commentary in the main results section. [...] I recommend that any theorem/corollary/proposition that is the primary basis for a claim made in the conclusion should receive significantly more commentary in Section 3."
>
> We agree, and we will work on improving the readability of that section.
>
> * "In the first paragraph of the conclusion, it says that "... the absence of ... is a rather general property ...".  The word "rather" should be clarified. "
>
> In that sentence, the word "rather" is included because one needs to assume that loss function (\ell) is smooth and convex. We will clarify this point in the conclusion and in Section 3.

---

### Official Review · AnonReviewer2 · 2019-10-23
**Official Blind Review #2**

**Rating:** 3

**Review:**

This paper studies the critical locus of loss functions of deep linear networks. More specifically, this paper introduces a distinction between pure critical points and spurious critical points. The authors prove that all non-global local minima are always pure for convex losses and provide a precise description of the number of topologically connected components of the global minima.

Generally speaking, this paper is well written. However, it is a little bit messy in Section 3.2, where the authors present 4 propositions, one corollary and one remark in less than one page.

My main concern is the contribution and importance of this paper as the landscape of the training loss function of deep linear networks has been well studied. This paper indeed provides some new techniques and insights in this research direction, but the authors do not discuss the possible extension to a more complicated and practical case. I would like to raise my score if the authors can provide some reasonable and convincing discussions regarding the extension to the nonlinear case.

Besides, there is no comparison with the existing results. The properties of critical points of deep linear networks have been studied in many literatures, the authors should provide detailed comparison and discussion between the derived results and these related work in the surrounding text of theorems.

 Another question is whether this analysis can be generalized to more practical settings, such as deep nonlinear networks. Besides, I believe that the critical points of general deep linear network can have an exact mapping to the critical points of normalized deep linear networks. Again, it is more interesting to explore the benefit of reparameterization for nonlinear networks.

The last question is whether the landscape analysis in this paper can shed some light on the training dynamics. This paper gives some results regarding the connected/disconnected components of global minima. Is it possible to show which global minima are more likely to be found by optimization algorithms?




**Experience Assessment:**

I have published one or two papers in this area.

**Review Assessment: Checking Correctness Of Derivations And Theory:**

I assessed the sensibility of the derivations and theory.

**Review Assessment: Checking Correctness Of Experiments:**

N/A

**Review Assessment: Thoroughness In Paper Reading:**

I read the paper at least twice and used my best judgement in assessing the paper.

---

> ### Author Response · Authors · 2019-11-11
> **Author Response to Review #2**
>
> Thank you again for the comments, in addition to our general response given above, we reply here to some specific concerns:
>
> * "it is a little bit messy in Section 3.2, where the authors present 4 propositions, one corollary and one remark in less than one page."
>
> We agree, and we will add some narration in between the results to improve the readability of that section.
>
> * "My main concern is the contribution and importance of this paper as the landscape of the training loss function of deep linear networks has been well studied. "
>
> Linear networks have been indeed studied in other papers, however we believe that previous works did not paint a complete picture of the loss landscape. For example, they failed to explain that the absence of non-global minima for linear networks holds in two very distinct settings: for the quadratic loss, it is true for any choice of widths due to special geometric reasons; for other smooth convex losses, it is only true in the case of "filling" architectures, because the functional space becomes convex. Furthermore, our analysis of linear networks is not the only contribution of our work, and indeed we introduce notions that can be applied in more general settings as well. We refer to our reply above for a detailed discussion.
>
> * "This paper indeed provides some new techniques and insights in this research direction, but the authors do not discuss the possible extension to a more complicated and practical case."
>
> We discuss this issue in our reply above. If the paper is accepted, we will include those observations in the paper as well.
>
> * "Besides, there is no comparison with the existing results. ... the authors should provide detailed comparison and discussion between the derived results"
>
> We agree, however it is often difficult to compare results, since we use a different language and our setting is more geometric. For example, other papers do not mention determinantal varieties (which correspond to the intrinsic functional space of linear networks) but these appear in all of our statements. However, we will provide a more detailed description of other works, highlighting whenever possible analogies and differences with our results.
>
> * "The last question is whether the landscape analysis in this paper can shed some light on the training dynamics. This paper gives some results regarding the connected/disconnected components of global minima. Is it possible to show which global minima are more likely to be found by optimization algorithms?"
>
> As discussed in the introduction, the focus of our paper is on "static" properties of linear networks (i.e., properties of the critical locus), which we believe to be a prerequisite to the investigation of training dynamics. However, we mention that Corollary 8 has an interesting consequence: if training is initialized below the energy level of the zero function then, assuming the dynamics converge, they must converge to a global minimum of the loss function. We are not sure whether this simple fact is known, but it is not explicitly stated in recent works that study the training dynamics of linear networks (e.g., in "Gradient descent aligns the layers of deep linear networks" by Ji and Telgarsky). Finally, regarding the structure of global minima, it seems reasonable to try and characterize the basins of attraction of the different connected components, as suggested by the reviewer. This seems like an interesting problem to which we currently don't know the answer (we suspect it may be related to the stratified structure of the singular locus of determinantal varieties).

---

### Official Review · AnonReviewer1 · 2019-10-29
**Official Blind Review #1**

**Rating:** 3

**Review:**

This paper studied the landscape of linear neural networks using algebraic geometry tools. They introduced a distinction between "pure" and "spurious" critical points. They showed that for quadratic loss, there are no spurious local minimum in both the filling and non-filling case. For other convex loss, there are no spurious local minimum in the filling case, but there are spurious local minimum in the non-filling case. They gave a precise description of the number of topologically connected components of the variety of global minima.

My concern to this paper is that the landscape of linear neural networks, which is the subject of this paper, has already been analyzed in the literature. The final results of this paper, though derived using a different approach, are not new. Another limitation of this paper is, the approach of algebraic geometry used in the analysis seems hard to be generalized to non-linear multi-layers neural networks.

A contribution of this paper is that the viewpoint of pure and spurious critical points made the landscape results of linear neural networks more intuitive. However, I don't have the expertise to assess whether this viewpoint was implicitly contained in the proof of previous results. Given this, I am not sure the contribution of this paper is enough for ICLR.

I don't hold a strong opinion, since there could potentially be great ideas inside the algebraic geometry tools.


**Experience Assessment:**

I have read many papers in this area.

**Review Assessment: Checking Correctness Of Derivations And Theory:**

I assessed the sensibility of the derivations and theory.

**Review Assessment: Checking Correctness Of Experiments:**

I did not assess the experiments.

**Review Assessment: Thoroughness In Paper Reading:**

I made a quick assessment of this paper.

---

> ### Author Response · Authors · 2019-11-11
> **Author Response to Review #1**
>
> Thank you again for the comments. In addition to our general response given above, we reply here to some specific concerns:
>
> * "My concern to this paper is that the landscape of linear neural networks [...] has already been analyzed in the literature. The final results of this paper, though derived using a different approach, are not new."
>
> It is indeed true that linear networks have been studied in other papers, however we believe that previous works did not paint a complete picture of the loss landscape. For example, they failed to explain that the absence of non-global minima for linear networks holds in two very distinct settings: for the quadratic loss, it is true for any choice of widths due to special geometric reasons; for other smooth convex losses, it is only true in the case of "filling" architectures, because the functional space becomes convex. We also refer to our reply above for a discussion of the main contributions of our work.
>
> * "Another limitation of this paper is, the approach of algebraic geometry used in the analysis seems hard to be generalized to non-linear multi-layers neural networks. "
>
> While we use algebraic tools to describe determinantal varieties, many of the concepts and ideas we introduce can be applied for deep nonlinear networks as well. This is discussed in detail in our reply above.

---

### Author Response · Authors · 2019-11-11
**Author Response**

We thank all the reviewers for their useful comments. We address here some shared concerns regarding the main contributions of our work and the extension of our analysis to non-linear networks. We respond to individual reviews in separate comment threads.

**Main contributions of the paper**

*New notions*

* We introduce the concepts of pure and spurious critical points. We believe that they provide an intuitive and useful language for studying a central aspect in the theory of neural networks, namely the (over)parameterization of the functional space and its effect on the optimization landscape. The notions apply to any neural network architecture (see Section 2.1 of the paper and the discussion below).
* In the context of linear networks, we define "filling" and "non-filling" architectures. Similar ideas were used in other works but the distinction was not emphasized. Analogous notions can be defined for non-linear networks (see Section of A.3 the paper and the discussion below) and we intend to investigate them in the future.

If the paper is accepted, we will clearly explain that these new notions are not limited to linear networks.


*New results*

To our knowledge, the following results/facts are new:
* Theorem 5, that describes the exact number of connected components of the set of global minima (or any other level set).
* Propositions 6 and 7, that give a very precise and general description of the critical locus of the loss. They also imply the interesting Corollary 8, that states that in a reasonable setting all critical points are in fact global minima.
* Theorem 12, that is an extension of the theorem of Eckart-Young. Despite being related to a classical fact, we were not able to find a result in the literature with such a precise description of the landscape (classification of all critical points) for the low-rank matrix approximation problem.
* Proposition 14 on the critical locus of normalized networks.
* We believe we are the first to point out with an example that linear neural networks with smooth convex loss functions may have many bad local minima (it is probably often believed that linear networks *always* have no bad local minima).

Other technical results, such as Theorem 4, are quite general and may be useful in other settings.


*Reinterpretation of existing results and connections to algebraic geometry*

Our geometric analysis unifies many existing results on linear networks, while at the same time making them more intuitive (as pointed out by reviewer 1). For example, many properties of the optimization landscape of linear networks are captured by the following non-trivial property: non-global minima of the loss in parameter space must correspond to non-global minima of the loss in function space (see Proposition 10).

In general, we believe that systematizing and reinterpreting older results is a fundamental process for science, and lays the foundations for new research. Similarly, spelling out connections with other fields can bring new perspectives and ideas. In particular, applied algebraic geometry can provide a powerful new toolset in the study of neural networks. As examples of applications of algebraic geometry to neural networks, we mention that networks with polynomial activation functions, which are in fact the simplest nonlinear generalization of our setup, can still be studied with algebro-geometric methods, since their neuromanifolds are algebraic varieties. Even ReLU networks lie in the scope of (real) algebraic geometry, as their neuromanifolds are semi-algebraic sets.

---

> ### Author Response · Authors · 2019-11-11
> **Author Response (continued)**
>
> **Extensions of the analysis to nonlinear networks**
>
> We discuss possible extensions of our analysis to the case of nonlinear networks. If the paper is accepted, we will include some of these observations in the conclusions.
>
> First, we emphasize that the notions of pure and spurious critical points are well-defined for arbitrary networks (indeed, they are introduced in Section 2.1 in a very general setting), and they can be used to classify critical points of any loss function. In fact, we conjecture that a variation of Proposition 10 remains true even for non-linear networks, i.e., we believe that non-global local minima are always pure. Informally, this means that the overparameterization of the functional space can never introduce new local minima but only saddles, so local minima are always related to the intrinsic geometry of the functional space. Note that arbitrary overparameterizations of a manifold do not have this property. If true, this result would illustrate a fundamental characteristic of neural networks.
>
> Many of the general strategies used in our proofs can in principle be applied to arbitrary networks (in particular, they could be used to address the conjecture mentioned above). Our analysis is in fact largely based on the study of the rank of the differential of the mapping from parameter to function space (Lemma 3 and Theorem 4), so this approach is potentially very general. In the case of linear networks, we made use of some known properties of determinantal varieties, namely their dimension and the characterization of their tangent space (see Section A1) to describe when the parameterizing map is a submersion. In the case of the functional spaces associated with arbitrary networks, similar characterizations would have to be derived from scratch, but should be feasible.
>
> Finally, the concept of "filling architecture" generalizes immediately to any situation where the functional space of a network is naturally embedded in a finite-dimensional ambient space. This is true for networks with polynomial activation functions. However, even if the activation is not a polynomial, it is possible to introduce a variation of the notion of filling architecture, by using an input dataset to map functions into a Euclidean space (this is explained in Section A.3). In this setting, the analogue of the conjecture mentioned above would have very important and general consequences for optimization. In particular, one could use the notion of "filling-ness" to give a precise statement on the conditions under which the landscape of the empirical risk does not have non-global minima, for any deep nonlinear architecture. This would significantly generalize similar results that have been shown in the case of shallow networks (see, e.g., Venturi et al. 18).

---

### Decision · Program_Chairs · 2019-12-19

**Decision:**

Accept (Poster)

**Comment:**

This paper studies the landscape of linear networks and its critical point. The authors utilize geometric properties of determinantal varieties to derive interesting results on the landscape of linear networks. The reviewers raised some concerns about the fact that many of the results stated here can already be achieved using other techniques and therefore had some concerns about the novelty of these results. The authors provided a detailed response addressing these concerns. One reviewer however still had some concerns about the novelty. My own understanding of the paper is that while some of these results can be obtained using other approaches the proof techniques (brining ideas from algebraic geometry) is novel and could be rather useful. While at this point it is not clear that the techniques generalize to the nonlinear case I think algebraic geometry perspective have a good potential and provide some diversity in the theoretical techniques. As a result I recommend acceptance if possible.